# Mutual repression between JNK/AP-1 and JAK/STAT stratifies senescent and proliferative cell behaviors during tissue regeneration

Janhvi Jaiswal[1,2], Janine Egert[3], Raphael Engesser[4], Andrea Armengol Peyrotón[1], Liyne Nogay[1,5], Vanessa Weichselberger[1,2], Carlo Crucianelli[1], Isabelle Grass[1,6], Clemens Kreutz[3,7], Jens Timmer[4,6,7], Anne-Kathrin Classen[1,6,7] *

1 Hilde-Mangold-Haus, University of Freiburg, Freiburg, Germany, 2 Spemann Graduate School of Biology and Medicine (SGBM), University of Freiburg, Freiburg, Germany, 3 Institute of Medical Biometry and Statistics, Faculty of Medicine and Medical Center, University of Freiburg, Freiburg, Germany, 4 Institute of Physics and Freiburg Centre for Data Analysis and Modeling, University of Freiburg, Freiburg, Germany, 5 International Max Planck Research School for Immunobiology, Epigenetics, and Metabolism, Freiburg, Freiburg, Germany, 6 BIOSS Centre for Biological Signalling Studies, University of Freiburg, Freiburg, Germany, 7 CIBSS Centre for Integrative Biological Signalling Studies, University of Freiburg, Freiburg, Germany

* anne.classen@biologie.uni-freiburg.de

**Data Availability Statement:** All relevant data are within the paper and its Supporting Information files.

## Abstract

Epithelial repair relies on the activation of stress signaling pathways to coordinate tissue repair. Their deregulation is implicated in chronic wound and cancer pathologies. Using TNF-α/Eiger-mediated inflammatory damage to *Drosophila* imaginal discs, we investigate how spatial patterns of signaling pathways and repair behaviors arise. We find that Eiger expression, which drives JNK/AP-1 signaling, transiently arrests proliferation of cells in the wound center and is associated with activation of a senescence program. This includes production of the mitogenic ligands of the Upd family, which allows JNK/AP-1-signaling cells to act as paracrine organizers of regeneration. Surprisingly, JNK/AP-1 cell-autonomously suppress activation of Upd signaling via *Ptp61F* and Socs36E, both negative regulators of JAK/STAT signaling. As mitogenic JAK/STAT signaling is suppressed in JNK/AP-1-signaling cells at the center of tissue damage, compensatory proliferation occurs by paracrine activation of JAK/STAT in the wound periphery. Mathematical modelling suggests that cell-autonomous mutual repression between JNK/AP-1 and JAK/STAT is at the core of a regulatory network essential to spatially separate JNK/AP-1 and JAK/STAT signaling into bistable spatial domains associated with distinct cellular tasks. Such spatial stratification is essential for proper tissue repair, as coactivation of JNK/AP-1 and JAK/STAT in the same cells creates conflicting signals for cell cycle progression, leading to excess apoptosis of senescently stalled JNK/AP-1-signaling cells that organize the spatial field. Finally, we demonstrate that bistable separation of JNK/AP-1 and JAK/STAT drives bistable separation of senescent signaling and proliferative behaviors not only upon tissue damage, but also in *Ras^{V12}*, *scrib* tumors. Revealing this previously uncharacterized regulatory network between JNK/AP-1, JAK/STAT, and associated cell behaviors has important implications for our conceptual understanding of tissue repair, chronic wound pathologies, and tumor microenvironments.

**Funding:** Funding for this work was provided by the Deutsche Forschungsgemeinschaft (DFG, German Research Foundation) under Germanýs Excellence Strategy (CIBSS – EXC-2189), the Emmy Noether Programme (CL490/1-1), the Heisenberg Program (CL490/3-1, CL490/4-1), the SFB850 / A08, and by the Boehringer Ingelheim Foundation (BIF Plus3) to AKC. The funders had no role in study design, data collection and analysis, decision to publish, or preparation of the manuscript.

**Competing interests:** The authors have declared that no competing interests exist.

**Abbreviations:** AED, after egg deposition; *egr*, *eiger*; Nub, Nubbin; ODE, ordinary differential equation; Rn, Rotund; ROI, region of interest; ROS, reactive oxygen species; SASP, senescence-associated secretory phenotype; SA-β-gal, senescence-associated β-galactosidase; *trbl*, *tribbles*; Upd, Unpaired; UPR, unfolded protein response.

## Introduction

Upon tissue damage, wound-derived factors initiate signaling pathways which drive cellular responses like apoptosis, proliferation, survival and tissue remodeling [1–4]. While these responses are essential for tissue repair, it is critical that they remain spatio-temporally restricted to avoid pathological consequences, such as the establishment of chronic wounds [5–7]. Chronic wounds are characterized by sustained inflammation, deregulated proliferation, and apoptosis, which are also hallmarks of tumors [8–11]. Different experimental models have identified the signaling pathways that control wound repair and tissue regeneration, many of which also drive diseases like cancer. Specifically, the JNK/AP-1 and the JAK/STAT pathways are consistently implicated in regeneration [12–15] and tumor growth [16–21]. In *Drosophila*, JNK/AP-1 is one of the earliest pathways activated in response to damage and indispensable for wound healing and regeneration [22–26]. In this role, JNK/AP-1 regulates a variety of conflicting cell behaviors including apoptosis [27,28], survival [29], and compensatory proliferation [30,31]. These paradoxical behaviors have been extensively characterized individually [32–34]. Yet, how they are spatially organized on a larger tissue scale to ensure regeneration is less well understood.

At the level of a single cell, these paradoxical cell behaviors are integrated by cell cycle progression. Using *Drosophila* wing imaginal discs (**S1A Fig**), we recently reported that high JNK/AP-1 signaling in wounds and tumors induces a cell-autonomous arrest or stalling of cells in the G2-phase of the cell cycle. G2-stalling is mediated by the JNK-dependent up-regulation of the G2/M pseudokinase *tribbles (trbl)* and the down-regulation of the G2/M phosphatase *string (stg/cdc25)*. Knock-down of *tribbles* or ectopic expression of *string* is sufficient to suppress G2-stalling and forces these cells into G1. This causes a substantial increase in apoptosis, which suggests that the G2-arrest protects high JNK-signaling cells from cell death and that reentering the cell cycle in wound environments is associated with cell lethality [29]. This model is supported by the idea that wounds produce reactive oxygen species (ROS); which can exacerbate cellular damage and cell death [26,35]. However, the molecular damage sensor p53 is specifically activated by the G2/M kinase Cdk1 to induce competence for damage-driven apoptosis only upon exit from G2 (**S1A' Fig**) [36]. Importantly, JNK/AP-1 signaling cells also produce mitogens belonging to the Unpaired (Upd) family [37–41], which activate the pro-proliferative JAK/STAT pathway essential for compensatory proliferation [16,18,26,42–51]. Paradoxically, this reveals that a JNK-signaling cell population exists at the center of wounds, which produces pro-proliferative signals but itself is prohibited from proliferation. These findings highlight the necessity for mechanisms, which ensure that high JNK/AP-1-signaling cells do not cycle despite being in the presence of mitogenic signals and that JAK/STAT-driven compensatory proliferation occurs only in cells that do not experience active JNK/AP-1-signaling. The regulatory network and molecular effectors that resolve this paradox have not been described. As JNK/AP-1 and JAK/STAT drive contradictory yet critical cell behaviors, characterizing their regulatory interactions is essential to understand how these pathways cooperatively organize tissue stress responses.

A stress-induced cell cycle arrest is often associated with senescence. Senescence was originally defined as the irreversible cessation of cell proliferation due to age, cellular damage, or oncogenic signaling and was suggested to drive age-dependent decline in regenerative capacity [52–56]. Senescent cells often exhibit a complex senescence-associated secretory phenotype (SASP) [2,57–59], which is linked to persistent production of inflammatory paracrine molecules and secretion of ECM degrading enzymes. However, senescent cells also exhibit up-regulation of autophagy, unfolded protein response (UPR), and ROS [2,57–59]. Importantly, senescent cells remain viable for long periods of time, indicating that they are resistant to

apoptosis, despite extensive cellular damage signals. However, recent studies reveal that the senescence program may also act transiently during wound healing of the mouse epidermis, where it may promote wound closure and cell plasticity [60–62]. Our previous work also started to suggest that G2-arrested cells may display senescent features that allow them to act as a central driver of physiological and pathological wound healing processes and contribute to the oncogenic potential of the tumor microenvironment [29]. Yet the precise features of this cell population are not described and competition between senescent cell cycle stalling and regenerative proliferation was not explored.

## Results

### JNK/AP-1 induces a cell cycle shift with senescent features upon tissue damage

To characterize the behavior of JNK/AP-1 signaling cells at a wound site in more detail, we analyzed signaling dynamics and repair behaviors induced by targeted expression of the TNF-α homologue *eiger* (*egr*). In many organisms including humans, TNF-α up-regulation is linked to tissue inflammation in physiological and pathological settings. The *rn*-GAL4 driver is active in the wing imaginal disc pouch (**Fig 1A**). *Egr* expressed ectopically using *rn*-GAL4 was previously used as a model to genetically induce tissue damage and study regeneration (**Fig 1B**) [63]. *egr* activates the stress-signaling pathway JNK/AP-1 after only 7 h of expression, as assessed by the sensitive JNK/AP-1 reporter *TRE-RFP* (**Fig 1C and 1D**) [64]. *TRE-RFP* expression further increases after 14 h and 24 h of *egr*-expression (**Fig 1E–1H**). Expression of *egr* and activation of JNK/AP-1 is associated with apoptosis, loss of epithelial polarity, tissue barrier dysfunction, and inflammation (**Figs 1I–1N, S1B, and S1C**) [22,23,65]. To further characterize the link between JNK-signaling, cell cycle stalling, and potentially senescent features, we monitored coevolution of G2 stalling and senescence markers at 7, 14, and 24 h of *egr*-expression. We specifically analyzed the well-established senescent marker senescence-associated β-galactosidase (SA-β-gal), the fluorescent cell cycle reporter FUCCI, and S-phase-driven incorporation of the nucleotide analogue EdU into DNA. Strikingly, in the JNK-signaling domain, SA-β-gal activity increased (**Figs 1O–1T, S1D, and S1E**), whereas EdU incorporation and G1-FUCCI cells decreased (**Figs 1B', 1U–1Z, and S1F-S1I**). The first evidence of a cell cycle shift could already be discerned after 7 h of *egr*-expression (**Figs 1U, 1V, S1F, and S1G**). In contrast, increase of SA-β-gal can be detected after 14 h of *egr*-expression (**Fig 1Q and 1R**), consistent with senescent features arising during a prolonged cell cycle arrest. In addition to SA-β-gal, other senescent and inflammatory markers can be easily identified after 24 h of *egr*-expression using immunofluorescence or scRNA-seq analysis, such as up-regulation of Upd family cytokines other paracrine factors resembling SASP, tissue matrix metalloproteases, redox defenses, NF-κB up-regulation, or unfolded protein response (**Figs 2A–2R and S2A-S2K**) [6,50,66–71]. Combined, these data strongly support a model where *egr*-expression induces JNK-signaling and, consequently, a cellular state characterized by a stalled cell cycle in G2 and by inflammatory features normally associated with terminal senescence.

### JNK/AP-1 and JAK/STAT separate into distinct spatial domains with distinct proliferative potential

Expression of *egr* induces *upd1-3*, the pro-mitogenic cytokines for the JAK/STAT pathway (**S2 Fig**; [72]) [18,26,48,50]. Yet, high JNK-signaling cells in *egr*-expressing discs that up-regulated a transcriptional *upd3* reporter did not exhibit JAK/STAT activity, as assessed by the *10xStat92E-dGFP* reporter (**Fig 3A–3G**) [73,74]. This is in stark contrast to wing disc

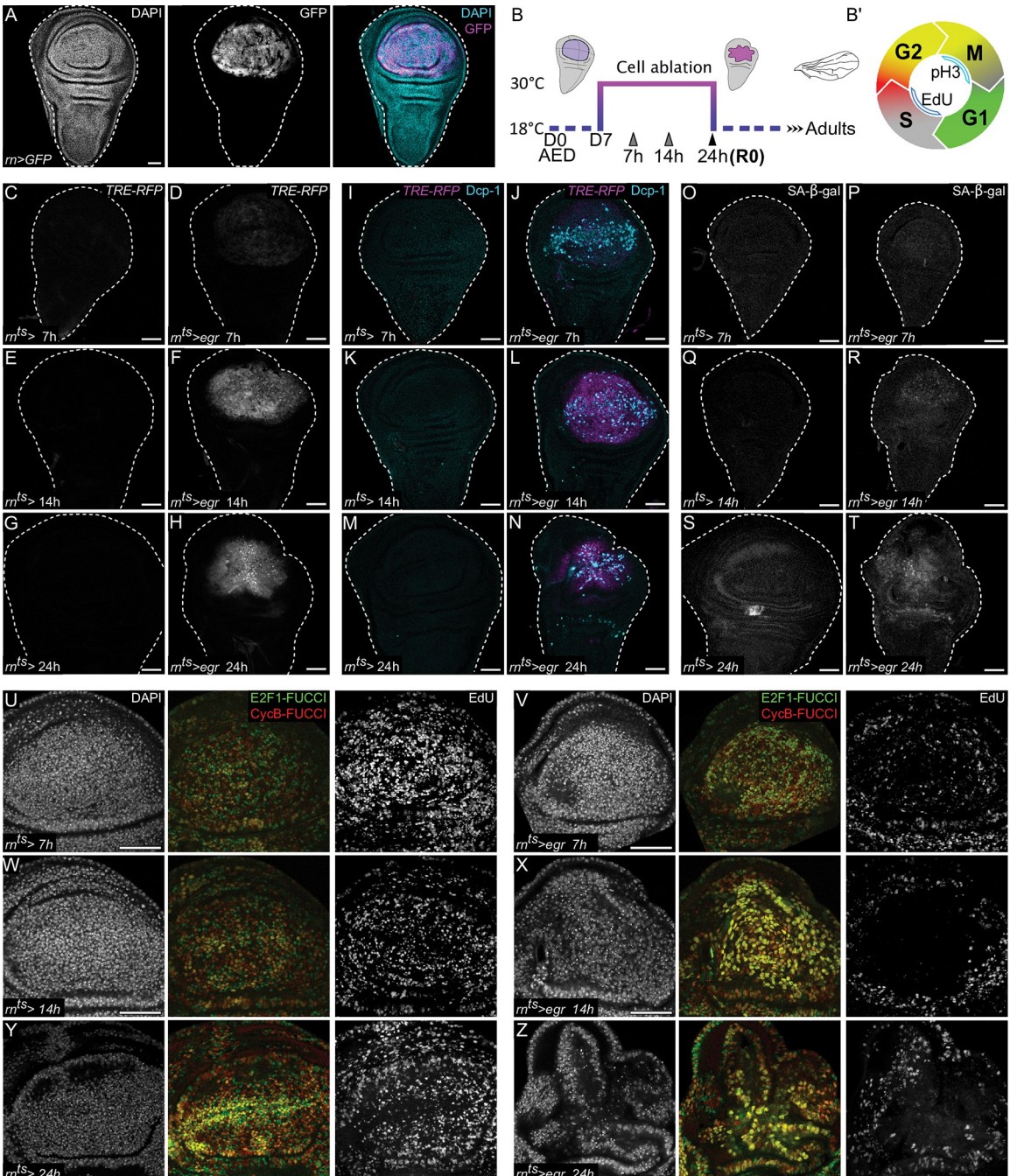

**Fig 1. *egr*-mediated tissue damage induces JNK/AP-1 signaling and cell cycle stalling in G2 with senescent features. (A)** A control wing disc after 24 h of *UAS-GFP*-expression in the pouch (R0), under the control of the *rn-GAL4* (*rotund-GAL4*) driver (*rn>*). *GFP* expression visualizes the pouch domain (magenta). **(B)** Experimental timeline denoting the protocol used for induction of *eiger* (*egr*)-expression in this study. Larvae were raised at 18˚C, and on day 7 after egg deposition (AED), vials were transferred to 30˚C to induce expression for 24 h (also referred to as recovery time point 0 or R0), using the pouch-specific *rotund-GAL4* (*rn-GAL4*) driver. Gray arrowheads represent time points used for time-course experiments. Shifting the larvae back to 18˚C terminates expression of *egr*. The adult wing blade arising from the damaged wing pouch allows for assessment of regenerative success. **(B')** Schematic of the FUCCI cell cycle indicator, utilizing the degradable *mRFP-NLS-CycB*$^{1-266}$ (red) and *GFP-E2F1*$^{1-230}$ (green) reporters as readouts for G1/late G2 and late S/G2 phases of the cell cycle, respectively [133,148]. Cells in the S-phase and M-phase are detected using EdU incorporation and pH3 staining, respectively. **(C-H)** A time-course analysis of the JNK/AP-1 reporter *TRE-RFP* in control discs and after 7 h **(C, D)**, 14 h **(E, F),** and 24 h **(G, H)** of *egr*-expression. A total of *n* = 26, 7 h and *n* = 39, 14 h control; and *n* = 33, 7 h *egr*-expressing and *n* = 21, 14 h *egr*-expressing discs were evaluated from *N* = 3 independent experiments. A total of *n* = 73, 24 h control and *n* = 92, 24 h *egr*-expressing discs were evaluated from *N* = 7 independent experiments. **(I-N)** A time-course analysis of apoptosis in response to egr-expression.

Discs were stained for cleaved Dcp-1 to visualize apoptotic cells (cyan) and express the JNK/AP-1 reporter *TRE-RFP* (magenta). Control discs and discs after 7 h **(I, J)**, 14 h **(K, L)**, and 24 h **(M, N)** of *egr*-expression are shown. A total of *n* = 20, 7 h control and *n* = 34, 7 h *egr*-expressing discs were evaluated from *N* = 2 independent experiments. A total of *n* = 32, 14 h control and *n* = 43, 14 h *egr*-expressing discs were evaluated from *N* = 3 independent experiments. A total of *n* = 63, 24 h control and *n* = 77 24h *egr*-expressing discs were evaluated from *N* = 6 independent experiments. **(O–T)** A time-course analysis of senescence-associated β-galactosidase (SA-β-gal) activity to detect senescent cells in control discs and *egr*-expressing discs after 7 h **(O, P)**, 14 h **(Q, R)**, and 24 h **(S, T)** of expression. A total of *n* = 10, 7 h and *n* = 14, 14 h control; and *n* = 16, 7 h *egr*-expressing and *n* = 15, 14 h *egr*-expressing discs were evaluated. A total of *n* = 23, 24 h control and *n* = 22, 24 h *egr*-expressing discs were evaluated from *N* = 2 independent experiments. **(U–Z)** A time-course analysis of the FUCCI cell cycle reporters $mRFP\text{-}NLSCycB^{1\text{-}266}$ (red) and $GFP\text{-}E2f1^{1\text{-}230}$ (green) and EdU incorporation to detect DNA replication activity in control discs and *egr*-expressing discs after 7 h **(U, V)**, 14 h **(W, X)**, and 24 h **(Y, Z)** of expression. A total of *n* = 18, 7 h and *n* = 20, 14 h control; and *n* = 6, 7 h *egr*-expressing and *n* = 10, 14 h *egr*-expressing discs were evaluated. A total of *n* = 10, 24 h control and *n* = 20, 24 h *egr*-expressing discs were evaluated from *N* = 4 independent experiments. Maximum projections of multiple confocal sections are shown for EdU staining in **U-Z**. Discs were stained with DAPI to visualize nuclei. Scale bars: 50 μm.

development where ligand expression and pathway activation patterns largely coincide and suggests that a specific regulatory network controls JAK/STAT activity upon tissue damage [74–77]. Consistent with paracrine activity of *upd1-3* [78,79], JAK/STAT signaling was broadly induced in the pouch periphery and hinge of *egr*-expressing discs (**Fig 3E**). Importantly, as proliferation was inhibited in the JNK-signaling domain at the wound center, proliferation was restricted to domains of JAK/STAT activation in the wound periphery (**Fig 3D–3G**).

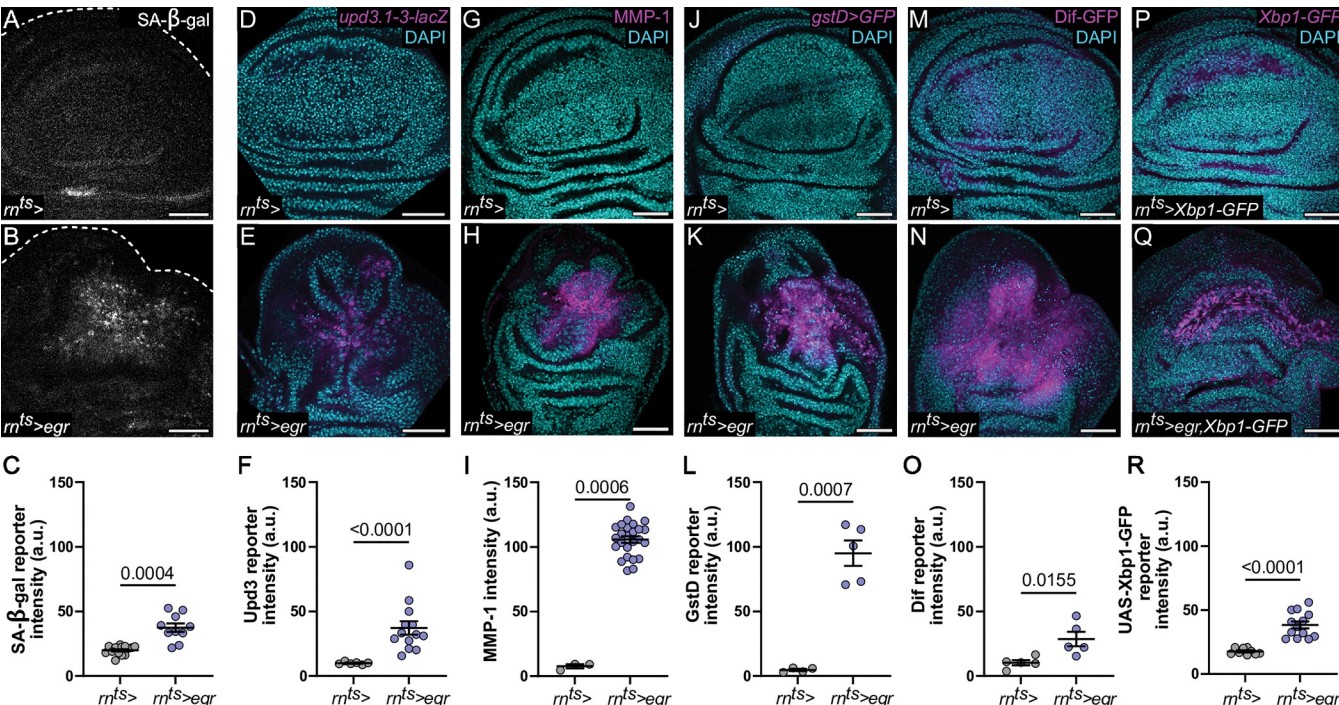

**Fig 2. JNK/AP-1 signaling cells display a senescence- and inflammation-associated signaling signature. (A-R)** Control **(A, D, G, J, M, P)** and *egr*-expressing **(B, E, H, K, N, Q)** discs after 24 h of expression (R0). *egr*-induced JNK/AP-1 signaling is associated with up-regulation of markers for inflammatory and senescent properties: SA-β-gal activity **(A, B)**, cytokines–*upd-lacZ* **(D, E)**, metalloproteases–MMP-1 **(G, H)**, ROS response–*gstD>GFP* **(J, K)**, UPR–spliced Xbp1-GFP **(M, N),** and NF-kB signaling–Dif-GFP **(P, Q)**. Fluorescence intensities of all marker were quantified in **(C, F, I, L, O, R)**. **(C)** Graph displays mean ± SEM for *n* = 15, control discs; *n* = 10, egr-expressing discs. Welch's *t* test was performed to test for statistical significance. **(F)** Graph displays mean ± SEM for *n* = 6, control discs; *n* = 13, egr-expressing discs. Mann–Whitney *U* test was performed to test for statistical significance. **(I)** Graph displays mean ± SEM for *n* = 3, control discs; *n* = 25, egr-expressing discs. Mann–Whitney *U* test was performed to test for statistical significance. **(L)** Graph displays mean ± SEM for *n* = 4, control discs; *n* = 5, egr-expressing discs. Welch's *t* test was performed to test for statistical significance. **(O)** Graph displays mean ± SEM for *n* = 5, control discs; *n* = 5, egr-expressing discs. Unpaired *t* test was performed to test for statistical significance. **(R)** Graph displays mean ± SEM for *n* = 9, control discs; *n* = 13, egr-expressing discs. Welch's *t* test was performed to test for statistical significance. Source data for quantifications provided in S1 File. Discs were stained with DAPI to visualize nuclei. Scale bars: 50 μm.

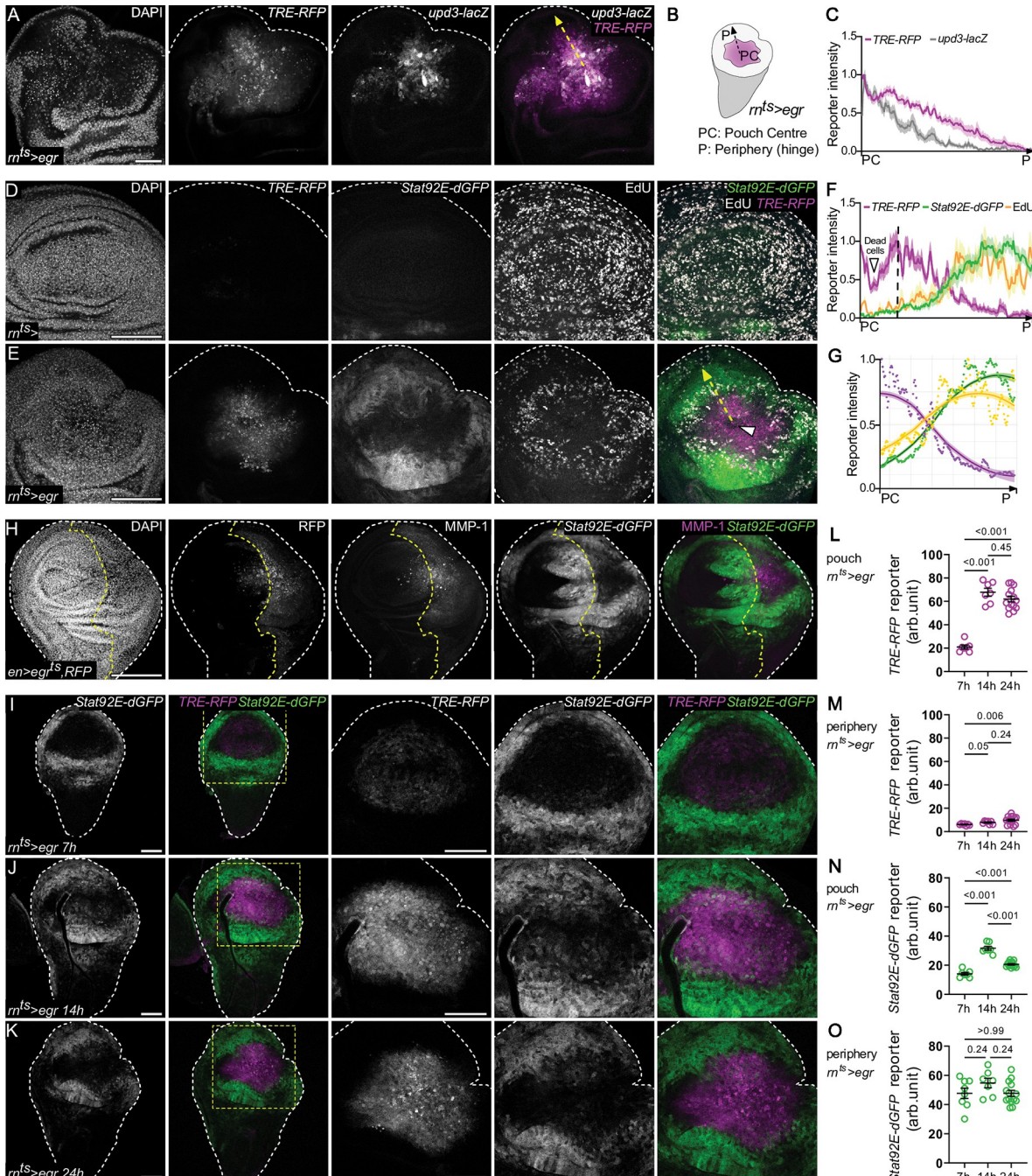

**Fig 3. JNK/AP-1 and JAK/STAT separate into distinct spatial domains with different proliferative potentials. (A)** *egr*-expressing disc also expressing the *upd3.1-lacZ* reporter (gray) and the JNK/AP-1 reporter *TRE-RFP* (magenta). Yellow arrow illustrates the principle of line traces used to generate graphs in (**C, F**). **(B)** Schematic representation of JNK/AP1 activity (magenta) after 24 h of *egr*-expression (R0). Arrow illustrates the vector of line traces used to generate graphs in (**C, F**). **(C)** Fluorescence intensity profiles for *upd3.1-lacZ* (gray) and *TRE-RFP* (magenta) reporters were traced along the axis from the pouch center (PC) to the disc periphery (DP, hinge) (arrows in **A, B**). Graph represents mean ± SEM of reporter fluorescence intensity values from *n* = 15 tracks across a representative disc, scaled to the maximum measured value. A total of *n* = 27, control and *n* = 33, *egr*-expressing discs were evaluated from *N* = 3 independent experiments. **(D, E)** A control **(D)** and *egr*-expressing disc **(E)** also expressing the *TRE-RFP* (magenta) and *Stat92E-dGFP* (green) reporters and assessed for S-phase activity using EdU incorporation assays. A total of *n* = 15, control and *n* = 19, *egr*-expressing discs were evaluated from *N* = 3 independent experiments. **(F)** Fluorescence intensity profiles for *TRE-RFP* (magenta), *Stat92E-dGFP* (green) reporters, and EdU (yellow) intensities traced from the pouch center (PC) to the disc periphery (DP) (black arrow in **B**). Graph represents mean ± SEM of reporter fluorescence intensity values from *n* = 15 tracks across a representative disc, scaled to the maximum measured value. **(G)** A smoothening function (see Materials and methods) applied to observed reporter patterns in (**F**) gives rise to a simple bistable pattern. Domain with dead

cells was excluded from the graph. **(H)** Expression of *egr* using the *en-GAL4* driver. The domain of expression in the posterior compartment was tracked using UAS-RFP, and efficiency of JNK-activation was assessed by staining for the JNK-target MMP-1 (magenta). Discs activate the JAK/STAT *Stat92E-dGFP* reporter (green) nonautonomously but not in the JNK-signaling domain. A total of *n* = 17, control and *n* = 17, *egr*-expressing discs were evaluated from *N* = 2 independent experiments. **(I-K)** A time-course analysis of *TRE-RFP* (magenta) and *Stat92E-dGFP* (green) reporter activity in *egr*-expressing discs after 7 h **(I)**, 14 h **(J)**, and 24 h **(K)** of expression. A total of *n* = 21, 7 h; *n* = 32, 14 h; and *n* = 31, 24 h control and *n* = 25, 7 h; *n* = 20, 14 h; and *n* = 25, 24 h *egr*-expressing discs were evaluated from *N* = 2 independent experiments. **(L-O)** Quantification of the fluorescence intensity of the JNK-reporter *TRE-RFP* **(L, M)** and the JAK/STAT reporter *Stat92E-dGFP* **(N, O)** within the JNK/AP1 signaling domain of the pouch ("pouch") and in a 20-μm band outside the JNK/AP1-signaling domain ("periphery") in *egr*-expressing disc (see also **S3W Fig**) after 7 h, 14 h, and 24 h of inductive temperature shift to 30˚C. Graphs displays mean ± SEM for *n* = 7, 7 h *egr*-expressing discs; *n* = 7, 14 h *egr*-expressing discs; and *n* = 14, 24 h *egr*-expressing discs. One-way ANOVA with Holm–Sidak's multiple comparisons test (periphery STAT) or a Brown–Forsythe and Welch ANOVA with Dunnett's T3 multiple comparisons test (other data sets) was performed to test for statistical significance. Source data for quantifications provided in S1 File. Maximum projections of multiple confocal sections are shown in **D, E, and H.** Discs were stained with DAPI (magenta) to visualize nuclei. Scale bars: 50 μm.

These observations highlight a pronounced spatial separation of signaling pathways and regenerative cell behaviors linked to JNK-induced tissue damage responses.

To further support this conclusion, we carried out a series of control experiments. We first wanted to rule out the possibility that the observed JAK/STAT activation pattern in the periphery of *egr*-expressing discs perpetuated from early hinge patterns of developing discs due to a Dilp8-induced developmental delay **(S3A and S3B Fig)** [80]. Hence, we monitored temporal changes in JAK/STAT activity via a time course of 0, 7, and 14 h of *egr*-expression under the control of *rn-GAL4*. Compared to developmental levels, we consistently observed ectopic induction of the JAK/STAT reporter in the hinge confirming that JAK/STAT activity is induced de novo in response to *egr*-expression **(S3A–S3M Fig)**. We also wanted to test if the same spatial separation of JNK/AP-1 and JAK/STAT activity can be induced by different patterns of *egr*-expression. We thus expressed *egr* in the posterior compartment of the wing disc using *en-GAL4* or in the hinge domain using *30A-GAL4*. To track the ability of these drivers to express *egr* and thus activate JNK, we monitored expression of the JNK/AP-1 target gene MMP-1 [81]. In both instances, cells that were expressing MMP-1 did not activate JAK/STAT signaling but JAK/STAT signaling was activated in the neighboring domains **(Figs 3H, S3N, and S3O)**. These results confirm that JAK/STAT activity can be induced nonautonomously and de novo in the hinge and pouch upon tissue damage. Yet, they also strongly indicate that cell-autonomous repression of JAK/STAT signaling may be an important consequence of JNK/AP-1 activation. This is consistent with recent observations of JNK/AP-1 and JAK/STAT separation in other contexts, such as epidermal wounds [12].

To gain more insight into the dynamics of spatial separation, we analyzed the temporal evolution of JNK/AP-1 and JAK/STAT activity in the JNK-signaling domain of control and *egr*-expressing discs. This analysis revealed that JAK/STAT was mildly elevated in the hinge (nonautonomously) and pouch (autonomously) after 7 h of *egr*-expression and even more by 14 h. At this point, many *egr*-expressing pouch cells displayed activation of either JNK/AP-1 or JAK/STAT reporter, or coexistence of both pathways. After 24 h of sustained *egr*-expression and high JNK/AP-1 activity, JAK/STAT activity remained high in the hinge periphery (nonautonomous activity) but had largely disappeared from the *egr*-expressing, JNK-signaling pouch cells (lack of autonomous activity) **(Figs 3I–3O and S3P-S3W)**. These observations suggest that short-term or moderate activity of JNK/AP-1 may facilitate local, intermixed activation of JAK/STAT. However, sustained or high JNK/AP-1 activity represses JAK/STAT activity cell autonomously. As a result, JNK/AP-1 and JAK/STAT signaling become separated into distinct cell populations within the tissue. Importantly, these patterns of JNK/AP-1 and JAK/STAT separation strongly correlate with spatial separation of regenerative cell behaviors, specifically senescent G2 stalling and compensatory proliferation, into spatially distinct signaling domains.

These data suggest that prolonged *egr*-expression and JNK activation organizes spatial patterns of damage-induced signaling and regenerative cell behaviors into a separated bistable field.

## JNK/AP-1 signaling represses JAK/STAT activity

Next, we wanted to understand if JNK/AP-1 signaling when activated by *egr*-expression can directly repress JAK/STAT activation cell-autonomously. We therefore analyzed clonal expression of a constitutively activated form of the JNKK Hep and coexpressed p35 to prevent cell death [82,83]. As expected, *hep*$^{act}$ clones activated JAK/STAT nonautonomously in the pouch, hinge, or peripodium but, importantly, strongly repressed JAK/STAT activation cell-autonomously (**Fig 4A–4D**). When clones were placed within the developmentally activated JAK/STAT domain in the hinge [76], cell-autonomous repression could also be observed (**Fig 4E**). These experiments confirm that JNK/AP-1 has the ability to activate JAK/STAT nonautonomously and strongly demonstrate that JNK/AP-1 represses JAK/STAT signaling cell-autonomously.

In these mosaic experiments, central pouch cells displayed lower levels of JAK/STAT activation. This could be due to the expression of Nubbin (Nub) and Rotund (Rn), 2 transcription factors described to reduce developmental Stat92E activity in the pouch [76,84]. To first exclude the possibility that JNK/AP-1 represses JAK/STAT by inducing pouch-specific transcription factors [84], we assessed Nub and Rn levels within *hep*$^{act}$ clones. *hep*$^{act}$ clones did not ectopically induce Nub or Rn; in fact, Nub was distinctly down-regulated (**S4A–S4C Fig**).

To exclude that activation of apoptosis and cell death in JNK-signaling cells contributed to repression of JAK/STAT, we also analyzed *egr*-expressing discs heterozygous for the *rpr,hid, grim* deficiency *Df(3L)H99* (**Fig 4F–4K**) or coexpressing *p35* (**S4D and S4E Fig**). These genotypes are expected to inhibit caspase activity at initiator or effector levels, respectively. Yet, spatial separation of JNK/AP-1 and JAK/STAT signaling could still be observed. These data strongly suggest that the cell-autonomous repression of JAK/STAT by JNK/AP-1 is neither dependent on developmental mechanisms of JAK/STAT repression nor on JNK-activated cell death pathways.

Lastly, to test if activation of JNK/AP-1 by different methods also drives separation of JNK/AP-1 and JAK/STAT signaling domains, we examined wing discs with reduced function of the well-characterized tumor-suppressor gene *scrib* [85–87]. Expression of *scrib-RNAi* with or without p35 under the control of *rn*-GAL4 activated both JNK/AP-1 and JAK/STAT in the pouch as described previously [85–87]. However, both signaling pathways were activated in an almost mutually exclusive pattern (**Figs 4L, 4M, S4F, and S4G**). These experiments suggest that JNK-mediated repression of JAK/STAT and spatial separation of signaling domains may be a robust feature of JNK/AP-1-driven processes.

## JNK/AP-1 drives a mutual repression network, which organizes spatial patterns during tissue repair

Next, we wanted to better understand the type of regulatory network in which JNK/AP-1 and JAK/STAT form spatially distinct signaling domains in tissues like the wing disc epithelium [88–91]. Regulatory networks describe how components of different signaling pathways interact and are integrated to define a cell's signaling state and response [92–96]. How regulatory networks can organize cellular behaviors to influence tissue-level outcomes in development has been extensively described using mathematical modeling [97–99]. To characterize the rules of the regulatory network for JNK/AP-1 and JAK/STAT during tissue damage and repair responses, we developed a mathematical model using partial differential equations. These equations describe the spatiotemporal evolution of JNK/AP-1 and JAK/STAT activation, and

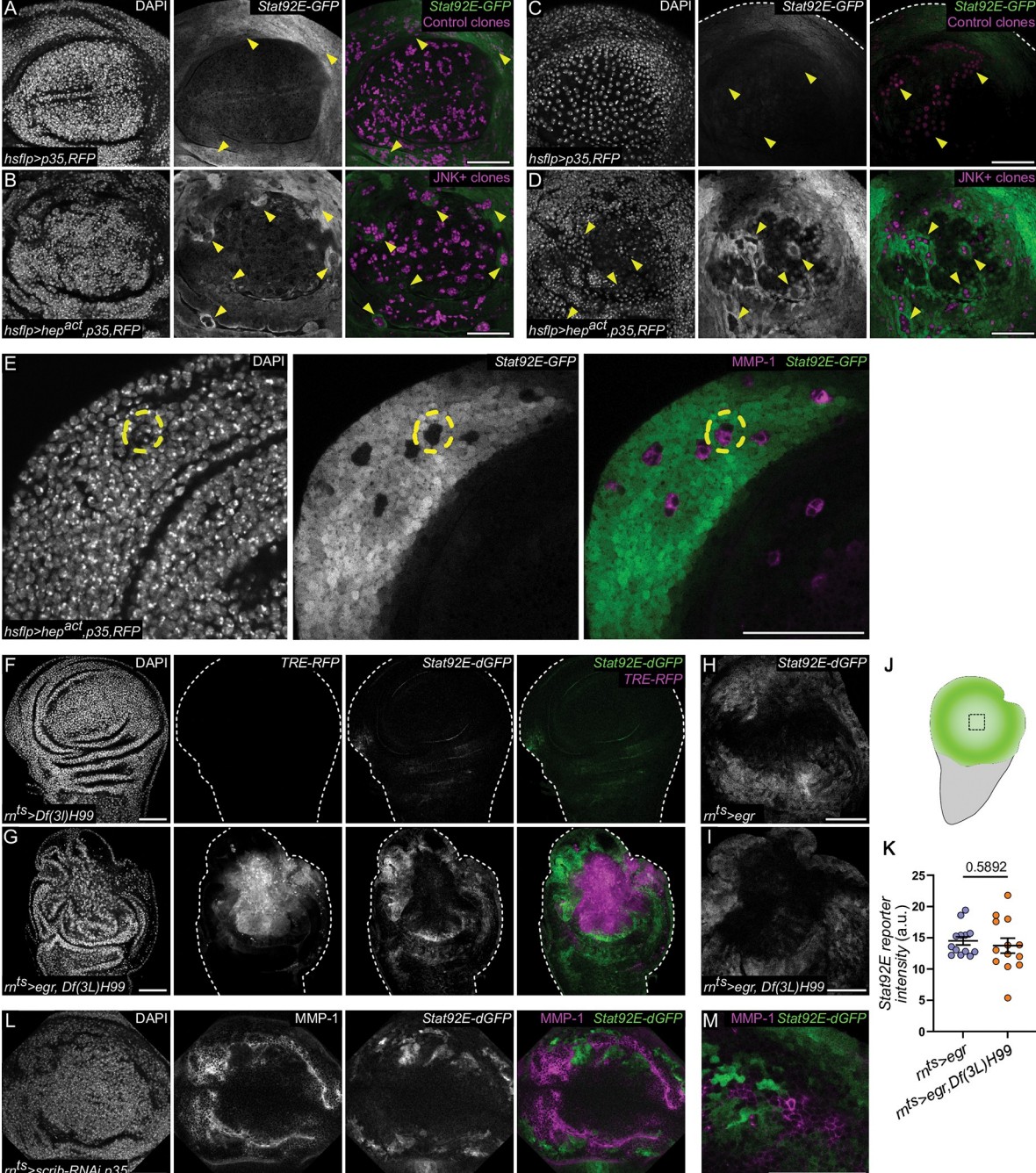

**Fig 4. JNK/AP-1 signaling represses JAK/STAT activity. (A-D)** *RFP* expression marks *p35*-expressing control clones in the pouch (magenta, **A, C**) or *p35,hep^act^*-coexpressing clones (JNK+) (magenta, **B, D**), in the peripodium (**A, B**) or in the pouch proper (**C, D**) at 28 h after clone induction in discs also expressing the stable *Stat92E-GFP* reporter (green). Yellow arrowheads indicate example clones of interest. Note the cell-autonomous repression of *Stat92E-GFP* reporter activity in JNK+ clones in the pouch and peripodium, when compared to control clones. Note the absence of non-cell-autonomous *Stat92E-GFP* reporter activity around control clones. A total of *n* = 8, control discs were evaluated from *N* = 2 independent experiments, and *n* = 13, *p35,hep^act^*-coexpressing discs were evaluated from *N* = 3 independent experiments. **(E)** *p35,hep^act^*-coexpressing clones in a disc expressing the stable *Stat92E-GFP* reporter (green) reporting developmental JAK/STAT activation patterns 48 h after clone induction. The disc was stained for MMP-1 (magenta)—a downstream target of the JNK/AP-1 signaling pathway to confirm pathway activation. The yellow circle highlights a clone, which shows strong cell-autonomous repression of the developmentally regulated JAK/STAT activity in the hinge. A total of *n* = 7, *p35,hep^act^*-coexpressing discs were evaluated. **(F, G)** A control **(F)** and *egr*-expressing disc **(G)** assessed for JAK/STAT reporter activity when caspase activity is reduced. Both discs a heterozygous for *Df(3L)H99*, a chromosomal deletion of *hid*, *rpr*, and *grim*, all central regulators of DIAP and thus caspase activity. Discs also express the *TRE-RFP* (magenta) and *Stat92E-dGFP* (green) reporters. **(H-K)** An *egr*-expressing control disc **(H)** and an

*egr*-expressing disc heterozygous for *Df(3L)H99* (**I**) expressing the *Stat92E-dGFP* (green) reporter. Quantification of the *Stat92E-dGFP* reporter fluorescence intensity measured within *egr*- and *egr, Df(3L)H99*-coexpressing central pouch domain. Black square in schematic (**J**) shows measured region. Graph (**K**) displays mean ± SEM for *n* = 13, *egr*-expressing discs and *n* = 13, *egr*-expressing disc heterozygous for *Df(3L)H99*. Unpaired *t* test was performed to test for statistical significance. (**L, M**) *scrib-RNAi, p35*-coexpressing discs stained for MMP-1 to detect JNK/AP-1 activity (magenta). Disc also expresses the *Stat92E-dGFP* (green) reporter. Note distinct cell-by-cell segregation and nonoverlapping patterns between the MMP-1 staining (magenta) and the *Stat92E-dGFP* (green) reporter. A total of *n* = 4, control and *n* = 6, *scrib-RNAi,p35*-coexpressing discs were evaluated. Source data for quantifications provided in S1 File. Discs were stained with DAPI (magenta) to visualize nuclei. Scale bars: 50 μm.

in particular, it allows the calculation of the steady state behavior for given initial concentrations of the signaling compounds [95,100]. We defined a set of experimentally determined rules to describe a unilateral repression model (**Fig 5A**), namely, (1) JNK/AP-1 effectors self-propagate JNK/AP-1 activation via production of paracrine factors, such as *egr* or diffusible ROS [26,101–103]; (2) JNK/AP-1 effectors activate JAK/STAT via production of paracrine factors, such as Upds [18,26,38,48,50,78,79,104,105]; (3) both pathways use positive feedback loops to enhance and stabilize their own activation [38,106–108]; and (4) JAK/STAT represses JNK/AP-1 cell-autonomously. This last condition is based on our previous work [50], which demonstrated that JAK/STAT is required to repress the self-propagating, nonautonomous expansion of JNK/AP-1 signaling within the imaginal disc (by mechanisms listed in point (1)) [50]. To simplify this unilateral repression model, we excluded negative feedback loops on JNK/AP-1 and JAK/STAT [109–111], as they often have more modulatory function in signal buffering or dynamics [112].

Based on our experimental findings that JNK/AP-1 repressed JAK/STAT cell-autonomously, we designed an additional model that, together with the aforementioned rules, included a rule that JNK/AP-1 represses JAK/STAT signaling (**Fig 5A**) [113]. As JNK and JAK/STAT repress each other, the model describes a so-called mutual repression motif. Next, we examined the unilateral and mutual repression models for their ability to recapitulate experimentally observed JNK/AP-1 and JAK/STAT patterns in *egr*-expressing discs. Using an unbiased sampling approach with over $10^6$ parameter sets, we compared the ability of the 2 models to reproduce the spatial patterns of JNK/AP-1 and JAK/STAT activity curves in *egr*-expressing discs by evaluating the correlation between the experimental curves and the spatial steady-state behavior of the mathematical model ("observed bistable pattern"). We found that the mutual repression model is more likely to reproduce the experimentally observed JAK/STAT patterns than the unidirectional model and to produce more "observed" bistable outcomes than the unidirectional model (**Figs 5B-5B" and S5A**). However, when we analyzed the model in more detail, we found that the correlation of the experimentally observed curves limited the spatial resolution of the model, as it does not provide possible solutions in a narrow JNK-activity space (**S5B Fig**). We thus reexamined the parameter space of the unilateral and mutual repression model using a set of descriptive features to characterize spatially separated patterns ("simple bistable pattern"); see Materials and methods for definition of descriptive features. Importantly, the mutual repression model performed again much better than the unilateral repression model in recapitulating the separated patterns, as substantially more parameter combinations generate the "simple" bistable patterns (**Fig 5C–5C"**). This suggests that the repression of JAK/STAT by JNK/AP-1 is a robust mechanism. Supporting this conclusion, mutual repression networks have been extensively described to drive the formation of bistable states [114–120]. We conclude that the mutual repression network lies at the core of JNK/AP-1 and JAK/STAT signaling interactions in response to tissue damage. Indeed, if JNK/AP-1 and JAK/STAT were not repressing each other cell-autonomously, JNK/AP-1 and JAK/STAT signaling would be expected to expand from the wound site in an unrestrained manner. Thus,

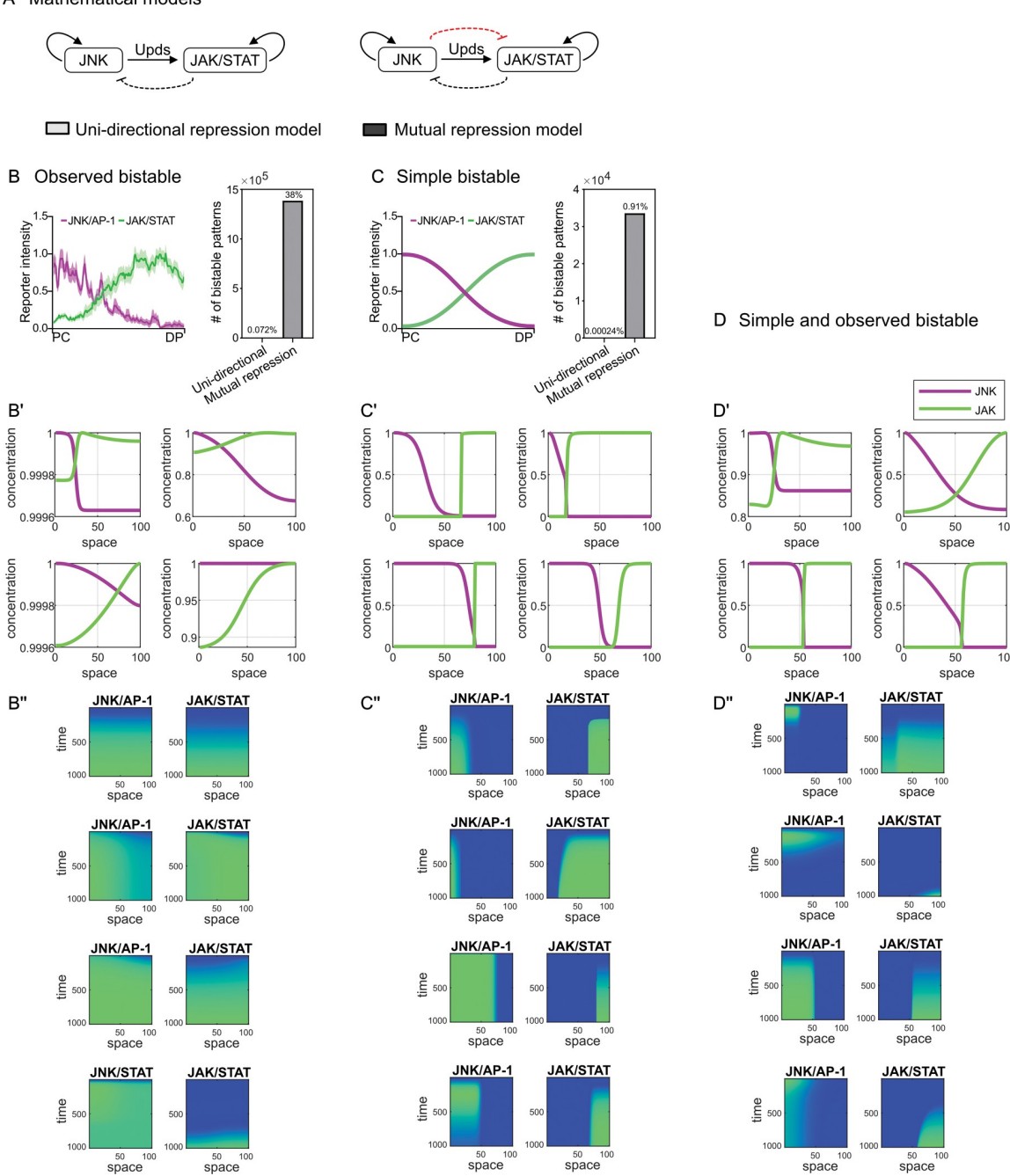

**Fig 5. Bistable pattern formation by JNK/AP-1 and JAK/STAT signaling by a mutual repression motif.** **(A)** Schematic representation of the mathematical models. In the unidirectional repression model only JAK/STAT inhibits JNK/AP-1 signaling, in the mutual repression model JNK/AP-1 and JAK/STAT signaling both inhibit each other. Please refer to the main text for details. **(B)** The steady-state patterns are classified as "observed" bistable patterns if the correlation of both the JNK/AP-1 activity and JAK/STAT activity curves with the experimentally observed curves is greater than 0.7. The number of "observed" bistable solutions obtained from the mutual repression model is substantially greater than that from the unidirectional repression model. Graph depicts number and percentage of bistable solutions obtained by searching more than $10^6$ sets of different parameters. Please refer to Materials and methods for details. **(B')** Representative steady-state simulation results for JNK/AP-1 and JAK/STAT signaling gradients classified as "observed" bistable patterns. **(C)** The steady-state patterns are classified as "simple" bistable when JAK/STAT decreases and JNK/AP-1 increases in space with a relative change greater than 10% (see Materials and methods for details). The number of "simple" bistable solutions obtained from the mutual repression model is substantially greater than that from the unidirectional repression model. Graph depicts number and percentage of bistable solutions obtained by searching more than $10^6$ sets of different parameters. Please refer to Materials and methods for details. **(C')** Representative steady-state simulation results for JNK/AP-1 and JAK/STAT signaling gradients classified

as "simple" bistable patterns. **(D, D')** Representative steady state simulation results for JNK/AP-1 and JAK/STAT signaling gradients classified as "observed and simple" bistable patterns. These solutions were identified when the descriptive features for "simple" bistable patterns were used to further filter positive solutions from "observed" bistable patterns. **(B", C", D")** Simulated spatiotemporal evolution of JNK/AP-1 and JAK/STAT activation for the same examples shown in **(B', C', D')**. These plots represent kymographs with a space and time axis, where pathway activation is plotted from low (blue) to high (green) activity. Source data for graphs derived from mathematical modeling provided in S2 File.

the mutual repression regulatory network also ensures that these signaling pathways remain restricted to the original site of damage.

An inspection of individual examples generated by the mutual repression model revealed intuitive spatial and temporal dynamics, even more so when we applied the descriptive features used to score for "simple" bistable patterns to further filter positive solution from "observed" bistable patterns (**Fig 5B'-5D' and 5B"-D"**). As the possible parameter space for any of these solutions was extremely large (**S5C–S5H Fig**), we were not able to narrow it down experimentally in this study. Yet, we wanted to explore, if specific parameter ranges are more important than others to establish a bistable signaling field. In developmental organizers, diffusion of paracrine factors is an important determinant of signaling patterns. We thus asked if concentrations of *egr* are important to determine the steady-state patterns by correlating *egr* concentrations with the position of the bistable intersection between JNK/AP-1 and JAK/STAT signaling. However, the size of the *egr* gradient was not associated with the position of intersection (**S5I Fig**). Rather JNK/AP-1 activity correlated strongly with the position of the intersection point but was unrelated to *egr* concentrations (**S5J and S5K Fig**). This suggests that it is not *egr* concentration but its interpretation into JNK activity is important for the emergence of bistable patterns. Indeed, we observed that the number of bistable solutions produced by the model increased with increasing rates of JNK/AP-1 activity (**S5L Fig**).

Taken together, we propose that JNK/AP-1 initiates a mutual repression network between JNK/AP-1 and JAK/STAT signaling that, via the production of Upds, facilitates the spatial separation of these 2 pathways. Thus, induction of JNK/AP-1 signaling upon tissue damage and mutual repression with JAK/STAT is sufficient to set up temporal and spatial signaling patterns and is likely required to establish functionally distinct cell populations during tissue repair and regeneration. The strong similarities with paracrine signaling centers regulating spatial patterning networks linked to specific cell behaviors and fate during development encourages us to propose that the JNK-induced mutual repression network with JAK/STAT functions as a central wound organizer network [121,122].

### *Ptp61F* and Socs36E repress JAK/STAT activity in JNK/AP-1 signaling cells

To understand how JNK/AP-1 may cell-autonomously repress JAK/STAT activation, we first asked if JNK/AP-1 down-regulates core components of the JAK/STAT pathway. Our previous study demonstrated that the core components *dome*, *hop*, and *Stat92E* [123] were not transcriptionally altered in *egr*-expressing discs [50]. Supporting this observation, we found that the expression of GFP-tagged Hop and Stat92E from validated lines were unaltered in *egr*-expressing cells (**S6A–S6I Fig**) [124]. This suggests that repression of JAK/STAT was due to other regulators of JAK/STAT signaling. However, levels of the negative JAK/STAT regulators dPIAS/Su(var)2-10 [125] and Apontic [126] were unchanged, but *ken* expression [127] was elevated in *egr*-expressing discs (**S6J-S6L' Fig**). However, neither knock-down of *ken* nor *dPIAS/Su(var)2-10* increased JAK/STAT activity in *egr*-expressing cells. Similarly, we could not reproduce findings that knock-down of *apontic* reinstated JAK/STAT signaling in *egr*-expressing discs (**S6M–S6O" Fig**) [128].

We then turned to the tyrosine phosphatase Ptp61F and the SOCS-box domain protein Socs36E—both of which are known repressors of Stat92E activation [129–131]. *Ptp61F* and *Socs36E* transcripts are significantly up-regulated in a cell cluster, which is characterized by senescent and JNK-signaling signatures, in a scRNA-seq analysis of *egr*-expressing discs (**S6P and S6Q Fig**) [72]. We found that knock-down of *Ptp61F* or of *Socs36E*—using RNAi lines that also derepress developmental JAK/STAT activity (**S6R–S6W Fig**)—caused up-regulation of Stat92E reporter activity in high JNK-signaling cells of *egr*-expressing discs (**Fig 6A–6H**). This suggests that Ptp61F and Socs36E are specifically required in JNK-signaling cells to suppress Stat92E activity. To furthermore test if levels of Stat92E in JNK-signaling cells affect JAK/STAT activity, we overexpressed a functional HA-tagged Stat92E (**S6X and S6Y Fig**) [132] in *egr*-expressing cells. Indeed, *egr,Stat92E-HA*-coexpressing cells in the central pouch domain displayed higher levels of JAK/STAT reporter activity, when compared to *egr*-expressing discs (**Fig 6I–6M**). Importantly, this demonstrates that signal transduction from Upd ligands down to the effector Stat92E is, in principle, active in JNK/AP-1-signaling cells. It is therefore the ability of negative JAK/STAT regulators like Ptp61F and Socs36E to limit Stat92E activation or promote its deactivation, which is important for JNK/AP-1-driven repression of JAK/STAT.

## JAK/STAT repression is required for cell cycle stalling in G2 and to protect arrested cells from apoptosis

As spatial separation of JNK/AP-1 and JAK/STAT signaling domains was a robust feature of *egr*-expressing discs, we wondered if it was necessary for regeneration. We thus asked if ectopic reactivation of JAK/STAT altered the behavior of JNK/AP-1 signaling cells. We reactivated JAK/STAT signaling in *egr*-expressing cells by overexpression of *Stat92E*, and by knock-down of *Ptp61F* or *Socs36E*, and closely monitored cell and tissue level responses, such as cell survival and proliferation. Reactivation of JAK/STAT signaling in all 3 *egr*-expressing genotypes led to a pronounced increase in apoptosis in the tissue (**Fig 7A–7I**). Expression of *UAS-GFP* in *egr*, *Stat92E*-coexpressing cells revealed that apoptotic cells originated from this coexpressing cell population (**S7A–S7D Fig**), demonstrating that JNK/AP-1 and JAK/STAT coactivation in the same cell is detrimental for cell survival.

We hypothesized that the pro-proliferative function of JAK/STAT interfered with the G2 cell cycle arrest of JNK/AP-1-signaling cells. Previous work demonstrated that genetically overriding the G2-arrest and forcing JNK/AP-1-signaling cells to cycle increases the probability of these cells to undergo apoptosis, due to activation of p53 by G2/M kinases (**see S1A' Fig**) [29,36]. We thus analyzed the ability of Stat92E to override the cell cycle arrest in the G2-phase. In undamaged control discs, targeted expression of *Stat92E* using *rn*-GAL4 altered the proportion of cells in S-phase consistent with its reported role in cell cycle acceleration [133], yet importantly, it did not alter the proportion of cells in G1 or G2 (**S7E–S7H Fig**). In contrast, a cell cycle analysis of *egr,Stat92E-HA* coexpressing domains revealed a substantial increase in G1-phase cells, indicating that JNK-signaling cells did not stall in G2 anymore (**Fig 7J–7M**). Moreover, cells undergoing active DNA replication could now be observed in the JNK/AP-1-signaling domain, indicating that JNK-signaling cells were actively cycling (**Fig 7N and 7O**). Knock-down of *Ptp61F* in *egr*-expressing discs also led to an increase in DNA-replicating cells (**Fig 7P and 7Q**). We conclude that forcing Stat92E activation in high JNK-signaling cells is sufficient to overcome the JNK-dependent G2 arrest, forcing cells to transition through G2/M into G1 and thereby increases the probability of these cells to undergo apoptosis.

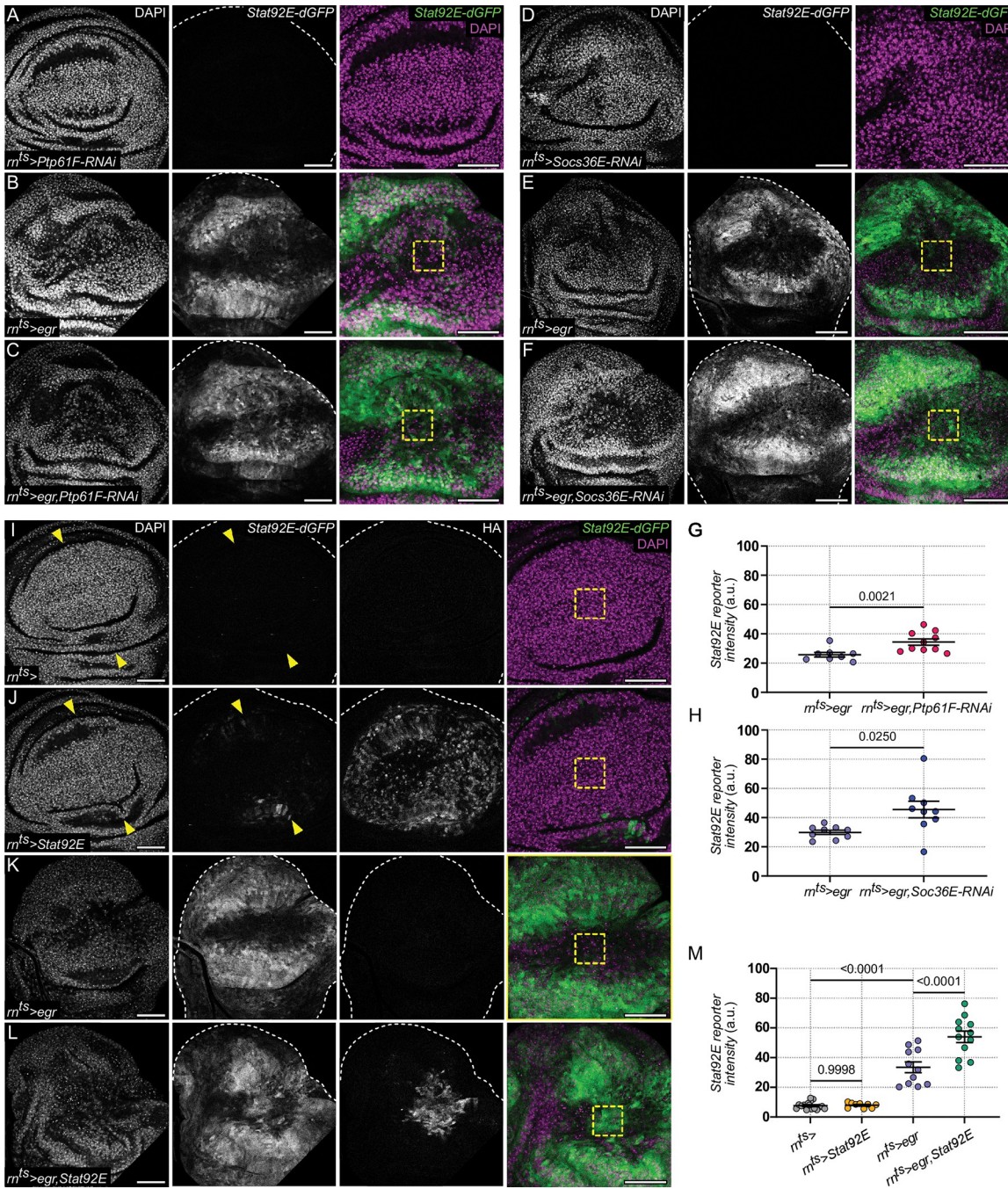

**Fig 6. *Ptp61F* and *Socs36E* repress JAK/STAT activity in JNK/AP-1-signaling cells. (A-F)** A *Ptp61F-RNAi*-expressing (**A**), *egr*-expressing (**B**)**,** and *egr,Ptp61F-RNAi*-coexpressing disc (**C**) after 24 h of expression. A *Socs36E-RNAi*-expressing (**D**), *egr*-expressing (**E**), and *egr,Socs36E-RNAi*-coexpressing disc (**F**) after 24 h of expression. All discs also express the dynamic *Stat92E-dGFP* reporter (green). Yellow squares highlight the central domain of remaining *rn-GAL4*-expressing cells and regions of measurement (see **Fig 4J**). The position and morphology of this central domain was characterized by us in previous experiments using coexpression of *UAS-GFP* constructs, which was incompatible with the use of the *Stat92E-dGFP* reporter. Note that derepression of JAK/STAT activity occurs in the central domain of *egr,Ptp61F-RNAi*- and *egr,Socs36E-RNAi*-coexpressing disc. **(G, H)** Quantification of the *Stat92E-dGFP* reporter fluorescence intensity measured within the central pouch domain of *egr*-expressing, *egr,Ptp61F*-RNAi-coexpressing discs (**G**) and *egr*, *Socs36E-RNAi*-coexpressing discs (**H**). Graphs display mean ± SEM for *n* = 8, *egr*-expressing discs; *n* = 10, *egr,Ptp61F-RNAi*-coexpressing discs; *n* = 9 *egr*-expressing discs; *n* = 9, *egr,Socs36E-RNAi*-coexpressing discs. Mann–Whitney *U* test (**G**) and Welch's *t* test (**H**) were performed to test for statistical significance. **(I-L)** A control (**I**), *Stat92E*-expressing (**J**), *egr*-expressing (**K**)**,** and *egr,Stat92E*-coexpressing disc (**L**) after 24 h of expression. All discs also express the dynamic *Stat92E-dGFP* reporter (green). Staining for HA confirms expression and nuclear/cytoplasmic localization of the *Stat92E-3xHA* construct. Yellow arrowheads point to mild JAK/STAT

up-regulation in the peripheral pouch of control discs (**I, J**). Yellow squares highlight the central domain of remaining *rn-GAL4*-expressing cells and regions of measurement. The position and morphology of this central domain was characterized by us in previous experiments using coexpression of *UAS-GFP* constructs, which was incompatible with the use of the *Stat92E-dGFP* reporter. Note that strong derepression of JAK/STAT activity occurs in the central domain of *egr,Stat92E*-coexpressing discs. (**M**) Quantification of the *Stat92E-dGFP* reporter fluorescence intensity measured within the central pouch domain for all genotypes shown in (**I-L**). Graphs display mean ± SEM for *n* = 16, control discs*; n* = 9, *Stat92E*-expressing discs; *n* = 11, *egr*-expressing discs; and *n* = 12, *egr*,*Stat92E*-coexpressing discs. One-way ANOVA with multiple comparisons was performed to test for statistical significance. Source data for quantifications provided in S1 File. Discs were stained with DAPI to visualize nuclei. Scale bars: 50 μm.

Our finding that the arrested G2-state could, in principle, be overcome was at odds with the cell population's senescent features, which would normally be associated with a terminally arrested cell cycle state. To understand if the JNK-induced cell cycle arrest is a transient cellular state, which can be reversed by the right signaling environment, we analyzed the proliferative potential of *egr*-expressing cells after *egr*-expression was terminated. Consistent with published lineage tracing results for this population [50], we found previously that *egr*-expressing cells started to proliferate within 48 h, which correlated with decreasing JNK/AP-1-signaling within the tissue (**Fig 7R and 7S**).

Our findings highlight the necessity of separating JNK/AP-1 and JAK/STAT signaling within the tissue. The JNK-induced mutual repression network ensures the establishment of 2 distinct and indispensable cell populations: (1) a JNK/AP-1-signaling cell population that stalls in G2, which prevents their apoptosis. Their function is to secrete necessary pro-mitogenic factors like Upds, or other paracrine effectors like Dilp8 and ImpL2 [18,80]; and (2) a JAK/STAT-signaling cell population that is able to respond to mitogenic signals and undergoes regenerative proliferation. This spatial separation would ensure that tissue damage sensing can be achieved by an apoptosis-resistant signaling center that survives wound-associated ROS or cellular damage and that regenerative proliferation can occur in an environment not exposed to wound-associated ROS, cellular damage, or inflammatory defense processes. In agreement with the idea that repression of JAK/STAT by JNK/AP-1 and thus bistable patterns are required for regeneration, we find that adult wings developing from *egr,Ptp61F-RNAi* coexpressing wing discs are smaller than adult wings from egr-expressing discs (**S7I Fig**).

## Bistable separation of JNK/AP-1 and JAK/STAT signaling during oncogenic growth

Our observation that tissue damage establishes a signaling-sending, G2-stalled cell population and a responding proliferative population led us to speculate that this patterning network could well support a tumor microenvironment. Interestingly, the coexistence of JNK/AP-1 and JAK/STAT pathways in tumors has been extensively described [16–21] and the existence of signal-sending G2-stalled cells controlling nonautonomous proliferation in tumors has been demonstrated [29]. However, their spatial patterns were not well defined.

To understand if JNK/AP-1 and JAK/STAT also engage in bistable patterning in tumors, we analyzed imaginal discs with reduced function of the well-characterized *scrib* tumor-suppressor gene [85] and ectopic expression of oncogenic $Ras^{V12}$ [134,135]. To generate a large genetically homogenous tissue, we again used the pouch specific *rn-GAL4* driver to express *scrib-RNAi* and $Ras^{V12}$ in the entire wing pouch. Oncogenic $Ras^{V12}$ expression for 44 h did not activate JNK/AP-1 or JAK/STAT reporters in the pouch (**Fig 8A and 8B**), demonstrating that $Ras^{V12}$ alone does not co-opt either pathway for growth. RNAi-driven knock-down of *scrib* induced barrier dysfunction (**S8A and S8B Fig**) and, consistent with previous reports [86,87], moderately activated both JNK/AP-1 and JAK/STAT in the pouch (please compare **S8C–S8F Fig**). A closer examination, however, revealed that the *scrib-RNAi* expressing pouch separated

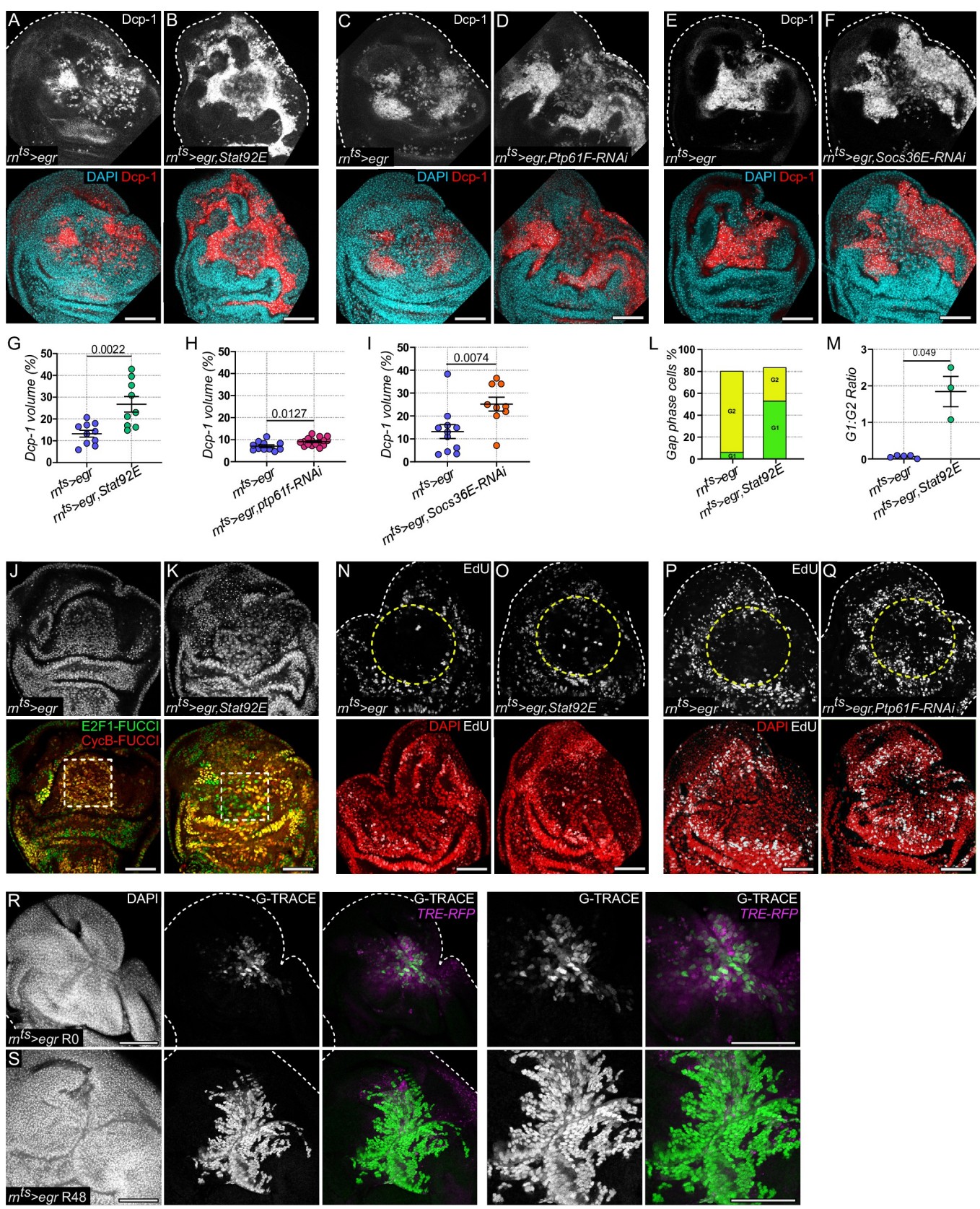

**Fig 7. JAK/STAT repression is required for cell cycle stalling in G2 and to protect arrested cells from apoptosis. (A-F)** *Egr*-expressing **(A, C, E)**, an *egr*, *Stat92E*-coexpressing **(B)**, an *egr,Ptp61F-RNAi*-coexpressing **(D)**, and an *egr,Socs36E-RNAi*-coexpressing **(F)** disc after 24 h of expression, stained for DAPI (cyan), and cleaved Dcp-1 (grey or red) to visualize apoptosis. **(G-I)** Quantification of percentage of volume occupied by cleaved Dcp-1 for all genotypes shown in **(A-F)**. Graph in **(G)** displays mean ± SEM for *n* = 10, *egr*-expressing and *n* = 9, *egr,Stat92E*-coexpressing discs. Unpaired *t* test was performed to test for statistical significance. Graph in **(H)** displays mean ± SEM for *n* = 11, *egr*-expressing and *n* = 16, *egr,Ptp61F-RNAi*-coexpressing discs. Unpaired *t* test was performed to test for statistical significance. Graph in **(I)** displays mean ± SEM for *n* = 11, *egr*-expressing and *n* = 9, *egr,Socs36E-RNAi*-coexpressing discs. Mann–Whitney *U* test was performed to test for statistical significance. **(J, K)** An *egr*-expressing **(J)** and *egr,Stat92E*-coexpressing disc **(K)** after 24 h of expression. Discs also express the FUCCI reporter. Dashed white squares highlight the central pouch domain of rn-GAL-expressing cells. Note the distinct shift from most cells in the G2-phase (yellow nuclei) to also cells in the G1-phase (green nuclei) within the central pouch domain. **(L, M)** Bar graph **(L)** representing the cumulative proportion of gap phase cells in G1 (green) and G2 (yellow) for each genotype. Graph **(M)** displays G1-phase:G2-phase ratios obtained from *n* = 5, *egr*-expressing and *n* = 3, *egr,Stat92E*-coexpressing discs. Dashed white squares in **(J, K)** indicate the position of the analyzed domain. Welch's *t* test was performed to test for statistical significance. **(N-Q)** EdU incorporation assays to detect DNA replication of cells in S-phase (gray) in *egr*-expressing **(N, P)**, or *egr*, *Stat92E*-coexpressing **(O)** and *egr,Ptp61F-RNAi*-coexpressing **(Q)** disc. Discs were stained with DAPI (red) to visualize nuclei. A total of *n* = 10, *egr*-expressing discs and *n* = 10, *egr,Stat92E*-coexpressing discs were evaluated. A total of *n* = 10 *egr*-expressing discs and *n* = 10, *egr,Ptp61F-RNAi*-coexpressing discs were evaluated. **(R, S)** An *egr*-expressing disc after 24 h of expression at R0 **(R)** and at recovery time point R48h **(S)** coexpressing an inducible permanent lineage label for *rn-GAL4*-expressing cells using the G-TRACE system (gray or green). Discs also express the *TRE-RFP* reporter (magenta) and were stained with DAPI to visualize nuclei. Note how the surviving rn-GAL4 population increases in size suggesting that stalled cells have started cycling again. A total of *n* = 8, *egr*-expressing discs at R0 and *n* = 9, *egr*-expressing discs at R48h were evaluated. Source data for quantifications provided in S1 File. Maximum projections of multiple confocal sections are shown in A-F and N-S. Scale bars: 50 μm.

JNK/AP-1 and JAK/STAT signaling, while only few cells displayed coactivation of both pathways (**Fig 8C**). This indicates that activation of JNK/AP-1 via disruption of cell polarity and barrier function also induces bistable separation of both pathways. Importantly, bistable separation also appears to act on short spatial scales between neighboring cells in a genetically homogenous tissue, which mirrors our results of short-term expression of *egr* for 7 h or 14 h in the pouch (see **Fig 3I and 3J** again).

When *Ras^{V12}* and *scrib* mutations occur in the same cell, they cooperate to cause dramatic overproliferation of imaginal discs, which is not observed in single *Ras^{V12}* or *scrib* mutant tissues. This cooperativity is thought to be driven by activation of JNK/AP-1, as a consequence of *scrib* disrupting cell polarity. JNK/AP-1 signaling then induces *upd's* and JAK/STAT activation, which can be utilized by *Ras^{V12}* to drive overgrowth [16,135]. We thus wondered if *Ras^{V12}* may disable the mutual repression motif between JNK/AP-1 and JAK/STAT. This would allow autocrine activation of pro-proliferative JAK/STAT by JNK signaling in the same cell, as generally depicted in the literature [16,135]. Cells driven to cycle despite activation of JNK/AP-1 may then be protected by antiapoptotic functions of *Ras^{V12}*. We wanted to test this hypothesis and closely monitored JNK/AP-1 and JAK/STAT signaling as well as cell proliferation in *Ras^{V12}* and *scrib-RNAi*-coexpressing discs. Importantly, we found that JNK/AP-1 and JAK/STAT activation were still clearly separated into exclusive spatial signaling domains (**Figs 8D–8F and S8G**). Moreover, a FUCCI cell cycle reporter for the G2-phase predominantly associated with JNK/AP-1 signaling cells, which repressed JAK/STAT cell-autonomously (**Figs 8G, 8H and S8H**). In contrast, the mitotic marker phospho-Histone H3 predominantly associated with JAK/STAT signaling cells (**Fig 8I–8K**). This suggests that even in a *Ras^{V12}* and *scrib* cooperativity model, where all cells are genetically identical, bistability driven by the mutual repression network may organize the tissue into distinct JNK/AP-1 or JAK/STAT-signaling domains with different proliferative functions. This result aligns with previous observations that cooperativity between *Ras* and *scrib* can rely, in principle, on nonautonomous interactions between JNK/AP-1 and JAK/STAT activated cells [16,136]. Crucially however, our observations demonstrate that nonautonomous cooperation between a JNK/AP-1-activating and a *Ras^{V12}* mutation may arise as a self-organizing principle from a mutual repression network, which is also active during tissue damage. As a consequence, distinct functional domains may emerge in a genetically homogenous tumorigenic tissue. Moreover, it also suggests that activation of this regulatory network and separation of senescent signaling from proliferative tasks

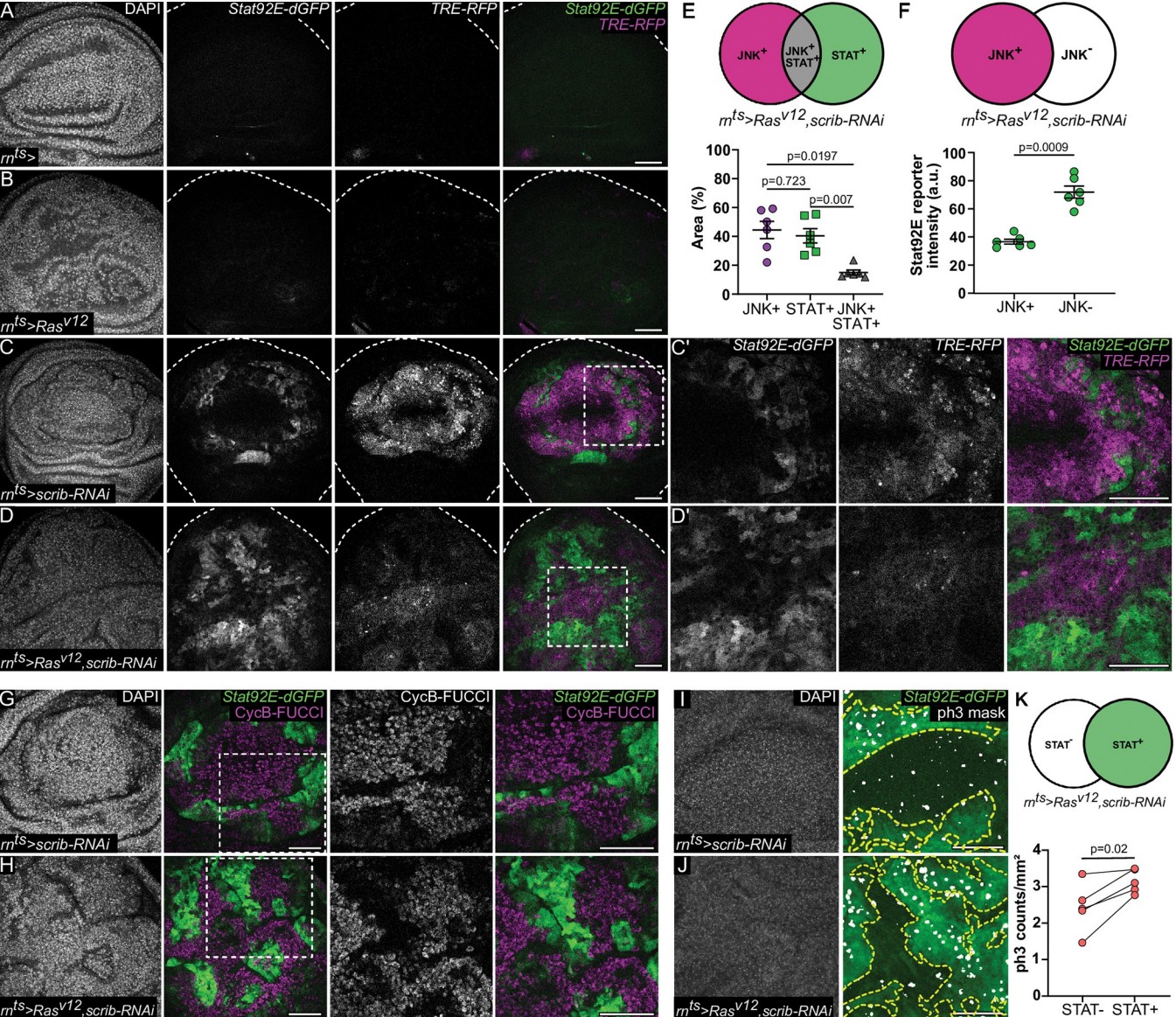

**Fig 8. Bistable separation of JNK/AP-1 and JAK/STAT signaling during oncogenic growth. (A-D')** A control wing disc (**A**) and wing discs expressing *Ras^v12* (**B**), *scrib-RNAi* (**C**), or *Ras^v12*,*scrib-RNAi* (**D**) for 44 h starting at D6 AED. Discs also express the JNK/AP-1 reporter *TRE-RFP* (magenta) and the JAK/STAT reporter *Stat92E-dGFP* (green). Dashed squares in (**C, D**) mark areas of higher magnification shown in (**C', D'**). A total of *n* = 10, control and *n* = 8, *Ras^v12*-expressing discs were evaluated from *N* = 2 independent experiments. A total of *n* = 21, *scrib-RNAi*-expressing and *n* = 27, *Ras^v12*,*scrib-RNAi*-expressing discs were evaluated from *N* = 3 independent experiments. (**E**) Schematic illustrating the *TRE-RFP*-positive (JNK+, magenta), *Stat92E-dGFP*-positive (STAT+, green), and JNK/AP-1 and JAK/STAT double positive regions (JNK+STAT+, gray) selected for measurements. Graph represents percentage of areas based on the total area analyzed. Note that only 15% of the tumor express both JNK/AP-1 and JAK/STAT signaling reporters. Graph represents mean ± SEM for *n* = 6, *Ras^v12*,*scrib-RNAi*-coexpressing discs. One-way paired ANOVA with Holm–Šídák's multiple comparison test was performed to test for statistical significance. (**F**) Schematic illustrating the *TRE-RFP*-positive (JNK+, magenta) and *TRE-RFP*-negative (JNK−, white) regions selected for measurements. Graph represents *Stat92E-dGFP* fluorescence intensity measured within selected JNK+ and JNK− regions (please see **S8G Fig** for *TRE-RFP* reporter intensities in JNK+ and JNK− regions). Graph represents mean ± SEM for *n* = 6, *Ras^v12*,*scrib-RNAi*-coexpressing discs. Paired *t* test was performed to test for statistical significance. (**G, H**) A wing disc expressing *scrib-RNAi* (**G**) and *Ras^v12*,*scrib-RNAi* (**H**) for 44 h. Discs also express the *Stat92E-dGFP* (green) reporter and FUCCI reporter *ubi-mRFP-NLS-CycB^1-266* (magenta), which is specifically expressed throughout the G2-phase of the cell cycle. Note how cells in G2 do not display JAK/STAT activation. A total of *n* = 3, *scrib-RNAi*-expressing and *n* = 4, *Ras^v12*,*scrib-RNAi*-expressing discs were evaluated. (**I, J**) A wing disc expressing *scrib-RNAi* (**I**) and *Ras^v12*,*scrib-RNAi* (**J**) for 44 h. Discs also express the *Stat92E-dGFP* reporter and were stained for phospho-Histone3 (ph3) to visualize cells in mitosis. Segmentation masks for ph3 were generated in FIJI for quantifications, and an overlay of the *Stat92E-dGFP* reporter (green) and the ph3 mask (white) is shown (right panel). Yellow dashed lines demarcate the STAT+ regions within the discs. A total of *n* = 7, *scrib-RNAi*-expressing and *n* = 12, *Ras^v12*,*scrib-RNAi*-expressing discs were evaluated. (**K**) Schematic illustrating *Stat92E-dGFP*-positive (STAT+, green) and *Stat92E-dGFP*-negative (STAT−, white) regions selected for measurements. Graph represents ph3 counts normalized to area in the STAT+ and STAT− regions for *n* = 5, *Ras^v12*,*scrib-RNAi*-coexpressing discs. Paired *t* test was performed to test for statistical significance. Source data for quantifications provided in S1 File. Discs were stained with DAPI to visualize nuclei. Scale bars: 50 μm.

may provide an advantage to tumors over autocrine integration of JNK/AP-1 and JAK/STAT signaling in the same cell.

## Discussion

Previous studies established a role for JNK/AP-1 in promoting regeneration upon injury [33,65]. JNK/AP-1 activates expression of *upds*, which are essential to promote proliferation and survival via JAK/STAT signaling [26,33,47,50,137–140]. By identifying a mutual repression network between JNK/AP-1 and JAK/STAT, we conceptually establish JNK as a core organizer of tissue repair with parallels to organizers of cell fate patterning in developing tissues. We demonstrate that crucial molecular mediators of JAK/STAT repression are Socs36E and *Ptp61F* downstream of JNK/AP-1. The rules of the regulatory network we describe have important implications. Previous reports suggest that JNK/AP-1 activates itself nonautonomously, for example, via activation of paracrine *egr* or ROS [26,102,103]. This implies that JNK/AP-1 activation could result in unchecked spatial expansion of JNK/AP-1-signaling, and consequently of JAK/STAT signaling, thereby disrupting the surrounding tissue and potentially causing pathological states, such as chronic wounds. However, the mutual repression motif restrains the expansion of both JNK/AP-1 and JAK/STAT signaling domains, thus defining stable regions of senescent, mitogenic signals and regenerative, proliferative responses for tissue repair (**Fig 9A and 9B**). Of note, our data also indicate that spatial patterns generated by the JNK/AP-1 and JAK/STAT regulatory network may act at different length scales, depending on the duration and strength of JNK/AP-1 activation. This could allow tissue repair processes to respond dynamically and facilitate continuous integration of the repair state. As wound healing progresses and the JNK/AP-1 signaling field shrinks, shifting spatial probabilities of senescent and cycling cells would ultimately resolve the damage. If a similar system may underlie spatial stratification of cell behaviors observed during repair of mouse epidermal wounds will be an important avenue for future studies [141].

At the level of cell behaviors, JNK/AP-1 and JAK/STAT generate mutually exclusive responses. JNK/AP-1 signaling supports wound behaviors like tissue sealing [22,23,81], cell fusion [12], establishment of paracrine signaling [72], and, importantly, an apoptosis-resistant state that depends on a G2 cell cycle arrest [29]. In contrast, JAK/STAT signaling supports compensatory proliferation, potentially via up-regulation of G1-S cyclins such as CycD or CycE [142,143]. Reducing JAK/STAT signaling upon damage is therefore one of the ways whereby JNK/AP-1 can prevent cycling. As the G2-arrest protects cells from apoptosis, it allows them to survive in a cytotoxic tissue damage environment and ensures that they can act as signaling centers and wound organizer, which coordinate tissue repair behaviors (**Fig 9C**). Importantly, our observations indicate that, with exceptions [144], even successful tumors avoid coexistence of JNK/AP-1 and JAK/STAT signaling in the same cell as they do not profit from the conflicting cell-autonomous inputs on proliferation and survival decisions (**Fig 9D**). Instead, tumors may highjack the mutual repression network to chronically activate the SASP-related senescent properties of JNK-signaling arrested cells to provide paracrine mitogenic signals and, as a consequence, need to outsource proliferation to other cells [29]. As tumors persist, prolonged or high JNK/AP-1 activity may separate JAK/STAT signaling into larger bistable fields. Indeed, prolonged JNK/AP-1 activity induced by chronic expression of *egr* appears to drive more defined separation of JNK/AP-1 and JAK/STAT into large bistable domains during tissue damage. Stabilizing signal domains may prolong resolution of the damage and increase the risk for chronic wound pathologies. In fact, chronic wounds in human patients display striking separation of cellular responses. In a central inflammatory domain, cells acquire senescent phenotypes, arrest, and fail to proliferate. In the periphery, a

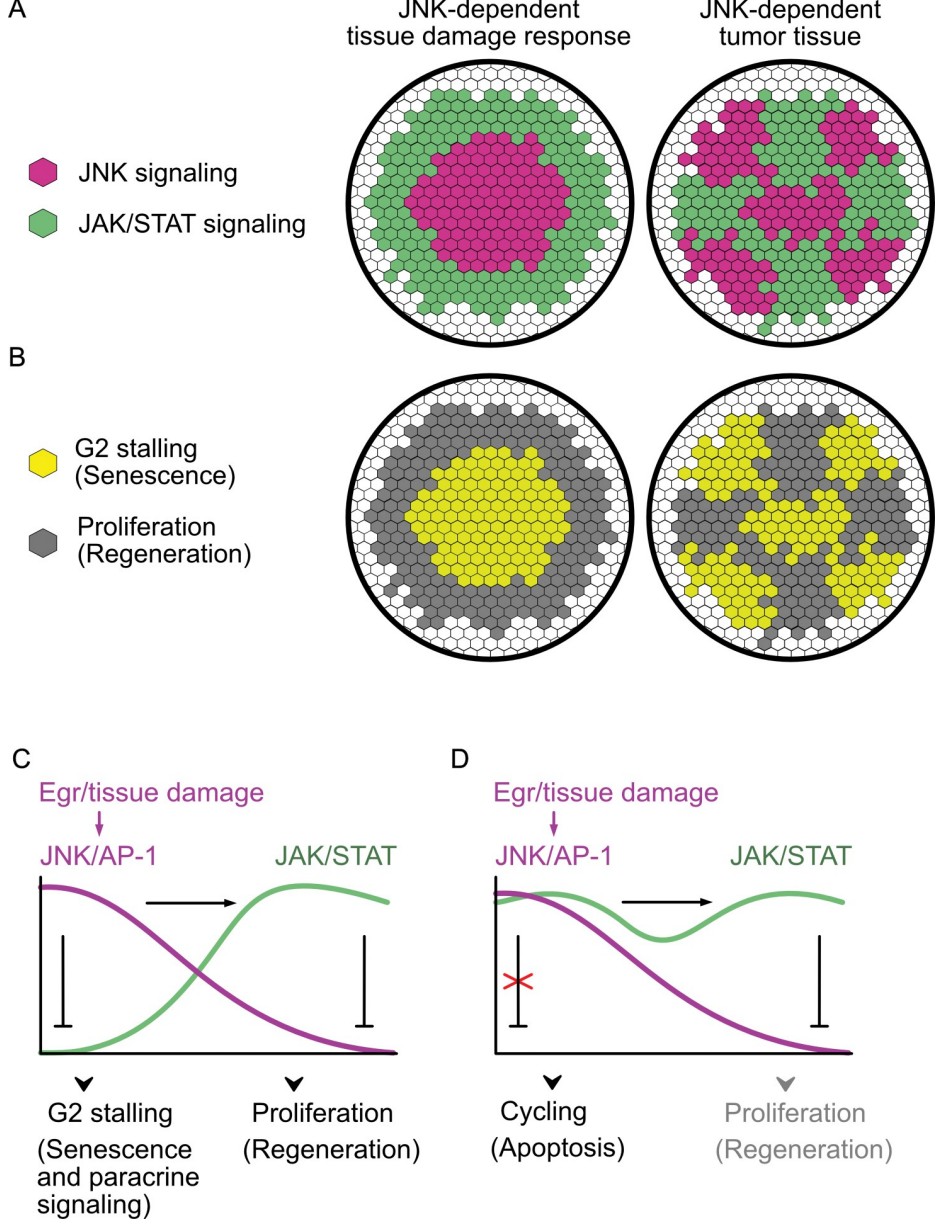

**Fig 9. Mutual repression between JNK/AP-1 and JAK/STAT sets up bistable separation of signaling domains, and thus senescent and proliferative behaviors. (A, B)** JNK-signaling induced by tissue damage or oncogenic transformation stratifies the tissue into distinct JNK/AP-1 and JAK/STAT signaling domains **(A)** and associated cell behaviors **(B)**. **(C)** JNK/AP1 and JAK/STAT signaling organize tissue repair via a mutual repression motif. JNK/AP-1 represses JAK/STAT activation, while JAK/STAT represses JNK/AP-1 activation. JNK signaling ensures G2 arrest and cell survival, while JAK/STAT signaling ensures proliferation and supports survival in the absence of JNK. **(D)** Disruption of the mutual repression motif by cell-autonomous activation of JNK/AP-1 and JAK/STAT signaling overrides the protective arrest in the G2-phase and drives cells through G2/M into G1, where p53 activation may increase the probability for damage-induced apoptosis.

hyperproliferative edge is evidence of chronic mitogenic signals emanating from the wound [1,7,8,145]. This spatial imbalance in proliferative activities mirrors those observed in *egr*-expressing discs, thereby suggesting that *Drosophila* imaginal discs may also represent a suitable model to study chronic wound healing pathologies [1,2,8].

## Materials and methods

### Fly stocks

All fly stocks and experimental crosses were maintained on standard media and raised at 18˚C or room temperature (22˚C) unless otherwise specified. For detailed genotypes, please refer to S1 and S2 Tables.

### Fly genetics

Fosmids carrying GFP-tagged *hop* (VDRC 318158) and *Stat92E* (BDSC 38670) alleles, expressed proteins at expected molecular weights and, at least partially, complemented respective null alleles, suggesting that both GFP-fusion proteins are functional. A potential Dome-GFP fosmid (VDRC 318098) failed to give rise to a GFP protein by immunofluorescence and western blot analysis and failed to complement a *dome* allele, thus Dome levels could not be directly analyzed. Genetic complementation assays were carried out by crossing the GFP-tagged fosmids into the background of LOF mutants (*dome$^{G0441}$*, *hop$^{9p5}$*, and *stat$^{85C9}$*). For X-linked *dome$^{G0441}$* and *hop$^{9p5}$* alleles, we calculated the percentage of hemizygous viable adult males emerging from control and experimental crosses. For the autosomal *stat$^{85C9}$* allele, we determined pupariation success of homozygous mutant larvae by counting the number of pupariated larvae in control and experimental crosses.

We validated RNAi lines by demonstrating that they caused changes to the developmental pattern of JAK/STAT activity in the hinge, if expressed throughout normal wing disc development. Crosses were set as previously described, using the *en-GAL4* driver to express the RNAi in the posterior compartment. Larvae were kept at 18˚C till D7 AEL. On D7, early L3 staged larvae were dissected, showing characteristic JAK/STAT reporter activity pattern in the hinge. Following dissection and staining, sibling controls and RNAi expressing discs were mounted on the same slide. All discs were imaged at the same settings and the JAK/STAT reporter intensities were quantified in selected regions in the anterior and posterior hinge. Intensity values from the anterior compartment (not expressing the RNAi) was used as the internal control.

### Flip-out clones

GAL4/UAS-driven "flip-out" overexpression experiments utilized heat shock–driven expression of a flippase to clonally express *a UAS* construct of choice. After 6 h of egg collections, heat shock was induced on developmental day 5 or 6 at 37˚C for 7 to 10 min. Larvae were dissected at wandering third instar stage or as indicated (28 h or 48 h after heat shock).

### Generating tissue damage and tumor models using the Gal4/UAS/Gal80(ts) system

To induce expression of *egr*, experiments were carried out as described in [29,42,50] with few modifications. Briefly, larvae of genotype *rn-GAL4, tub-GAL80$^{ts}$* and carrying the desired *UAS*-transgenes were staged with a 6-h egg collection and raised at 18˚C at a density of 50 larvae/vial. Overexpression of transgenes was induced by shifting the temperature to 30˚C for 24 h at D7 after egg deposition (AED), and larvae dissected at recovery time point (R0 h) unless noted otherwise. For time course and tumor experiments, transgenes were induced by shifting the temperature to 30˚C for 0 h, 7 h, 14 h, or 24 h at D7, or 44 h at D6, respectively. Larvae were subsequently dissected for analysis or allowed to recover at 22˚C for the indicated time. All images represent R0 h unless noted otherwise. Control genotypes were either *rn$^{ts}$>* or

sibling animals (+/*TM6B*, *tubGAL80 or* +/*TM6c*). All experiments were performed with $\geq 2$ independent experiments.

## Adult wings analysis and scoring

Crosses were allowed to develop to adulthood at appropriate temperatures. Adults were collected in fresh vials and scored within 12 h of eclosion. The scoring was carried out with the genotypes blinded, and wings were individually scored by visually assessing characteristic wing phenotypes and subsequently binned into 5 different classes: 0%, 25%, 50%, 75%, and 100% [29,42,50]. Any wings which were folded upon themselves were excluded from analysis. Only entirely undamaged wings were scored as 100%.

## Immunohistochemistry

Wing discs from third instar larvae were dissected and fixed for 15 min at room temperature in 4% paraformaldehyde in PBS. Washing steps were performed in PBS containing 0.1% TritonX-100 (PBT). Discs were then incubated with primary antibodies (described in **S1 Table**) in PBT, gently mixing overnight at 4°C. Tissues were counterstained with DAPI (0.25 ng/μl, Sigma, D9542), Phalloidin-Alexa Fluor 488/647 (1:100, Life Technologies), or Phalloidin-conjugated TRITC (1:400, Sigma) during incubation with cross-absorbed secondary antibodies coupled to Alexa Fluorophores (Invitrogen or Abcam) at room temperature for 2 h. Tissues were mounted using SlowFade Gold Antifade (Invitrogen, S36936). Whenever possible, experimental and control discs were processed in the same vial and mounted on the same slides to ensure comparability in staining between different genotypes. Images were acquired using the Leica TCS SP8 Microscope, using the same confocal settings and processed using tools in Fiji. Of note, all wing discs expressing the *10xStat92E-dGFP* reporter were boosted with an anti-GFP antibody.

## EdU labelling

EdU incorporation was performed using the Click-iT Plus EdU Alexa Fluor 647 Imaging Kit (described in **S1 Table**) prior to primary antibody incubation. Briefly, larval cuticles were inverted in Schneider's medium and incubated with EdU (10 μM final concentration) at room temperature for 15 min. Cuticles were then fixed in 4% PFA/PBS for 15 min and washed for 30 min in PBT 0.5%. EdU-Click-iT labeling was performed according to manufacturer's guidelines. Tissues were washed in PBT 0.1%, after which immunostainings, sample processing and imaging were carried out as described above.

## Senescence-associated β-galactosidase staining

CellEvent senescence detection kit from Invitrogen (C10850) was used to check SA-β-gal activity, following the instructions of manufacturer. Wing discs were dissected in PBS, fixed with 4% PFA, washed with 1% BSA (in PBS), and then incubated in working solution for 2 h at 37°C. Washing steps were performed in PBS and PBS containing 0.1% TritonX-100 (PBT). Tissues were counterstained with DAPI (0.25 ng/μl, 520 Sigma, D9542). Tissues were mounted using SlowFade Gold Antifade (Invitrogen, S36936).

## Western blots

Cell lysates from third instar wing imaginal discs and brains were prepared in lysis buffer (50 mM Tris-HCl (pH 7.5), 300 mM NaCl, 0.1 mM EDTA, 1% Triton X-100, 0.1% SDS, 5% Glycerol, 1 mM PMSF, 1/10 tablet of Complete Mini Protease Inhibitor Cocktail) on ice. Protein

samples were loaded onto a 10% polyacrylamide gel, along with Chameleon Duo Pre-stained Protein Ladder (LI-COR, P/N 928–60000) as a molecular weight ladder and run at 150 V. After SDS-PAGE, proteins were transferred onto a nitrocellulose membrane (Bio-Rad, 162–0115, 0.45 μm) in transfer buffer (25 mM Tris, 192 mM glycine, 20% (v/v) methanol) using the wet-tank method with a current density of 300 mA for 1 h. Prior to antibody incubation, membranes were blocked with 5% milk in PBS. Membrane was incubated with primary antibodies in PBS-T (1% Triton-X in 1xPBS)—rat anti-GFP (Chromotek, 3H9-100, 1:1,000) and mouse anti-α-tubulin (Sigma, T9026, 1:5,000) on a rotative plate at 4˚C overnight. Primary antibodies were removed and washed in dH$_2$0 twice for 5 min each. Membrane was then incubated with secondary antibodies diluted in PBS-T—donkey α-mouse secondary (LI-COR, 926–68072, 1:20,000) labelled with a 700-nm IRDye and goat α-rat (LI-COR, 926–32219, 1:20,000) labelled with an 800-nm IRDye. Membrane was washed in dH$_2$0 twice for 5 min each before detection using the Image Studio Software on the Odyssey SA system (LI-COR).

## Image analysis and quantification

**General comments.**  Images were processed, analyzed, and quantified using tools in Fiji (ImageJ v2.0.0) [146]. Extreme care was taken to apply consistent methods (i.e., number of projected sections, thresholding methods, processing) for image analysis. Figure panels were assembled using Affinity Designer v1.10. Statistical analyses were performed in Graphpad Prism or R v3.3.3 (www.R-project.org).

**Total disc and Dcp-1 volume quantification.**  Masks were generated across stacks after applying a fixed threshold to the DAPI channel (10–255) followed by "Fill holes" and "Despeckle" function In Fiji. A separate mask was similarly generated for the Dcp-1 channel (50–255) followed by the "Despeckle" function. The 2D area per section was obtained using the "setSlice" function to extract each z-section, followed by the "Measure" function. 2D area per section was then multiplied with the z-step size and summed for the entire z-stack for each disc to obtain the 3D volume. To control for differences arising from the size of the discs, ratios between the Dcp-1 volume and total disc volume (or DAPI volume) were obtained per disc and reported as percentage values.

**SASP marker quantifications.**  A central z-section comprising the senescent cell population (with large nuclei) was extracted from stacks using the DAPI channel as a reference. Region of interest (ROI) outlines were either manually drawn on the DAPI channel to include the pouch and hinge domains till the medial hinge fold. Centroid coordinates were identified using the "Measure" function. Fluorescence intensity of the SA-β-Gal, Upd3, Upd1, GstD, Dif, and Xbp1 reporters, or MMP-1 staining, was measured within a square ROI (25 × 25 μm) placed centrally over the centroid. Mean intensity within the ROI was obtained using the "Measure" function in Fiji.

***10xStat92E-dGFP*** quantification.  Maximum intensity projections of selected confocal sections were taken, excluding the peripodium and carefully chosen to capture signaling activity within the disc proper. In the *egr*-expressing or *egr,transgene*-coexpressing discs, a mask of the pouch domain was generated. ROI outlines were either manually drawn on the DAPI channel to include the pouch and hinge domains till the medial hinge fold or by thresholding Nub staining. Centroid coordinates were identified using the "Measure" function and marked with a point tool. To measure the fluorescence intensity of the *Stat92E-dGFP* reporter, a square ROI (25 × 25 μm) was placed centrally over the centroid, and mean intensity within the ROI was obtained using the "Measure" function in Fiji.

**Tracing JNK, JAK/STAT, EdU, and Upd reporter profiles.**  For reporter patterns along disc center to disc periphery, JNK/AP-1 masks were made using thresholding tools and ROI

outlines were generated. Based on ROIs, the centroid value was determined and $n \geq 15$ tracks were drawn from the centroid outwards towards the hinge for a fixed distance—covering the JNK/AP-1 domain, the JNK/AP-1-JAK/STAT interface, and the JAK/STAT signaling domain, respectively. Using the "Plot profile" function in Fiji, the fluorescence intensity values of each reporter was obtained along these tracks and averaged. Reporter patterns for each disc was graphed by scaling the averaged values from each track, between 0 and 1, and plotted on the Y-axis. This was done independently for $n \geq 3$ discs, and a representative image was selected for visualization of the spatial trends of the reporters.

**LOESS smoothing of traced reporter profiles.** Trend lines for JNK, JAK/STAT, and EdU reporter profiles were generated using the geom_smooth function of the ggplot2 visualization package in R (level = 0.95, alpha = 0.4, span = 0.8).

**Measuring the JNK/AP-1 and JAK/STAT reporter intensity within the pouch and periphery over time.** For time course experiments of spatial signaling patterns in *egr*-expressing discs, a central z-section was chosen. For control discs, a ROI was traced with line tools in Fiji using a Nub staining as a spatial reference for the maximal pouch domain. To closely approximate the rn-GAL4 domain, which is smaller than the Nub-positive domain by approximately 3 cell rows, the "Enlarge" function was used with a negative value ($-30$ pixel) to generate an ROI that covers only the Rn domain. Fluorescence intensity of the *TRE>RFP* (JNK) and *Stat92E-dGFP* (JAK/STAT) reporters was measured within this new ROI and outside of this ROI in a 15-μm broad band for 7 h, 14 h, and 24 h control discs. For egr-expressing discs, a ROI was defined as high *TRE>RFP* expressing domain equivalent to rn-GAL4 expressing cells. Fluorescence intensity of the *TRE>RFP* (JNK) and *Stat92E-dGFP* (JAK/STAT) reporters were measured within this new ROI and outside of this ROI in a 20-μm broad band for 7 h, 14 h, and 24 h control discs. Note that the hinge domain is slightly distorted and stretched in egr-expressing discs, thus a slightly larger band was used to capture a similar number of cells in the periphery.

**Tracing the JAK/STAT reporter profile in the hinge.** Partial max projections from confocal stacks of the disc proper were generated, carefully excluding any peripodial signal. For hinge profiles of the JAK/STAT reporter, size-matched discs were chosen, and line traces were manually drawn along the hinge using the polygon selection tool. Reporter intensity traces along the hinge were plotted using the "Plot profile" function for representative 0 h, 7 h, and 14 h control and *egr*-expressing discs.

**Cell cycle profiles.** Representative sections were selected per disc and a square ROI area ($75 \times 75$ μm$^2$) was placed over the central pouch cells. Only viable cells stained with characteristic euchromatic and heterochromatic DAPI staining were chosen. Using the "Multi point" tool, cells showing a G1 profile (green), G2 profile (yellow + red), or neither (black) were counted. Percentage values for the gap phases were calculated for $n \geq 3$ discs per genotype and visualized. $t$ Tests were performed on the calculated G1:G2 ratios per disc.

**JAK/STAT reporter quantifications for testing RNAi lines.** Crosses were set up as previously described, and wing discs from larvae were dissected on D7 AED. Single sections at similar focal planes were carefully chosen during imaging. Using drawing tools in Fiji, ROIs were generated to include JAK/STAT signaling cells in the hinge within A and P compartments. The mean JAK/STAT reporter intensity within the ROI was obtained from control and RNAi expressing discs using the "Measure" function in Fiji. For each dataset, paired $t$ tests were performed between A and P fluorescence intensity values for the same genotype.

**Nuclear translocation of Stat92E-GFP.** Images from the pouch ($n = 12$) and hinge ($n = 9$) domains of control discs and pouch ($n = 16$) and hinge ($n = 12$) domains of *p35+egr* expressing discs were obtained to track the intracellular localization of *Stat92E-GFP* within the nucleus and cytoplasm. Images were taken at high magnification (63×). After thresholding ("Huang"), the selection tool was used to generate a nuclear ROI using the DAPI channel. The inverse selection

tool was used to generate a corresponding cytoplasmic ROI for each image. These nuclear and cytoplamic ROIs were then placed on the *Stat92E-GFP* channel, and mean fluorescence intensity values were obtained for each subcellular fraction, and unpaired *t* tests were performed.

**JNK$^+$ and JNK$^-$ Area and intensity quantifications (*Ras$^{v12}$,scrib-RNAi* tumors).** A representative z-section was selected per disc. As our interest was in quantitating the spatial separation, only JNK/AP-1 or JAK/STAT signaling regions in the tumor were chosen. After applying a "Gaussian blur" filter (sigma = 2), "Moments" thresholding and "Despeckle" function for noise correction was applied to generate masks for regions showing *TRE>RFP* (JNK) and *Stat92E-dGFP* (JAK/STAT) reporter activity. From the "Image calculator" function, "AND" and "Subtract" operations on JNK/AP-1 and STAT masks were carried out to obtain the JNK$^+$STAT$^+$ mask, and the exclusive JNK$^+$ (JNK-STAT) or exclusive JNK$^-$ (STAT-JNK) masks, respectively. Area of each mask was obtained using the "Measure" function. Summed area of these masks were considered as the total area, and each area was then represented as a percentage of total. One-way paired ANOVA with Holm–Šídák's multiple comparison correction was performed to test for statistical significance. Mean intensity for *TRE>RFP* or *10xStat92E>dGFP* reporters were measured inside the JNK/AP-1 mask (JNK$^+$) and the exclusive JNK$^-$ masks. Paired *t* tests were performed to test for statistical significance.

**Mitotic index (ph3 counts).** Partial maximum intensity projection of 3 central z-sections were taken per disc. To generate masks for regions showing *Stat92E>dGFP* reporter activity (STAT$^+$), a "Gaussian blur" filter (sigma = 1), "Moments" threshold and "Despeckle" function for noise correction was applied. The DAPI channel was used to manually outline the pouch domain and generate masks. Using the "Image calculator" function, STAT$^+$ mask was "Subtract"-ed from the pouch mask to generate the STAT$^-$ mask. Areas of the STAT$^+$ and STAT$^-$ regions in the pouch were obtained using the "Measure" function. Then, the ph3 channel was duplicated and "Li" threshold was applied. "Remove outlier" (radius = 5 threshold = 50) and "Despeckle" function for noise correction, followed by "Watershed" function was applied to generate a ph3 mask. Using the "AND" operator, the ph3 mask was overlayed with the STAT$^+$ and STAT$^-$ masks. "3D object counter" function was then used to obtain ph3 "Object counts" within the STAT$^+$ and STAT$^-$ regions and divided by area of each ROI to get the ph3 counts per mm$^2$. Paired *t* tests were performed to test for statistical significance.

## Mathematical modelling

To test for the existence of a regulatory motif within our signaling network, a mathematical model was derived to describe the temporal dynamics of the concentration of specific molecules over a fixed 2D space. The partial differential equations were solved by discretizing space into 100 compartments and describing the dynamics in each compartment by a set of ordinary differential equations (ODEs), which are coupled between the compartments via diffusion of EIG and UPD (see also Fig 5). Two sets of modeling equations were established, that reflected networks which include or exclude repression of JAK/STAT by JNK. The term describing repression of JAK/STAT (referred to as "JAK" in the modeling equations and hereafter) by JNK/AP-1 is highlighted in red.

Unidirectional repression model:

$$\frac{dEIG(t,x)}{dt} = \mathrm{b}_{EIG}(x) - k_{deg,EIG}\,EIG + k_{act,EIG}\frac{JNK^{n1}}{K_{m,act,EIG}^{n1} + JNK^{n1}} + D_{EIG}\nabla^2 EIG$$

$$\frac{dJNK(t,x)}{dt} = \mathrm{b}_{JNK} - k_{deg,JNK}\,JNK + k_{act,JNK}\frac{EIG^{n2}}{(K_{m,act,JNK}^{n2} + EIG^{n2})(1 + k_{inh,JNK,by,JAK}^{ni1}\,JAK^{ni1})}$$

$$\frac{dJAK(t,x)}{dt} = b_{JAK} - k_{deg,JAK}\,JAK + k_{act,JAK}\,UPD\,\frac{JAK^{n3}}{K_{m,act,JAK}^{n3} + JAK^{n3}}$$

$$\frac{dUPD(t,x)}{dt} = b_{UPD} - k_{deg,UPD}\,UPD + k_{act,UPD}\,\frac{JNK^{n4}}{K_{m,act,UPD}^{n4} + JNK^{n4}} + D_{UPD}\nabla^2 UPD$$

Mutual repression model:

$$\frac{dEIG(t,x)}{dt} = b_{EIG}(x) - k_{deg,EIG}\,EIG + k_{act,EIG}\,\frac{JNK^{n1}}{K_{m,act,EIG}^{n1} + JNK^{n1}} + D_{EIG}\nabla^2 EIG$$

$$\frac{dJNK(t,x)}{dt} = b_{JNK} - k_{deg,JNK}\,JNK + k_{act,JNK}\,\frac{EIG^{n2}}{(K_{m,act,JNK}^{n2} + EIG^{n2})(1 + k_{inh,JNK,by,JAK}^{ni1}\,JAK^{ni1})}$$

$$\frac{dJAK(t,x)}{dt} = b_{JAK} - k_{deg,JAK}\,JAK + k_{act,JAK}\,UPD\,\frac{JAK^{n3}}{(K_{m,act,JAK}^{n3} + JAK^{n3})(1 + k_{inh,JAK,by,JNK}^{ni2}\,JNK^{ni2})}$$

$$\frac{dUPD(t,x)}{dt} = b_{UPD} - k_{deg,UPD}\,UPD + k_{act,UPD}\,\frac{JNK^{n4}}{K_{m,act,UPD}^{n4} + JNK^{n4}} + D_{UPD}\nabla^2 UPD$$

States (EIG, JNK, JAK, UPD), space (x), and time (t) are represented in characteristic units. The mathematical terms include parameters for the basal rate of production ($b$), activation rate ($k_{act}$), linear degradation rate ($k_{deg}$), Michaelis constant ($K_m$), inhibition constant ($k_{inh}$), diffusion coefficients ($D_{EIG}$, $D_{UPD}$), Hill coefficients to model cooperativity ($n_1, n_2, n_3, n_4, ni_1, ni_2$), and Hill kinetics to describe self-amplification and mutual repression.

The first equation describes the dynamics of the concentration of Egr, produced at a basal rate dependent on the spatial coordinate x and degraded at a linear rate. Hill kinetics describe the production of *egr* induced by JNK/AP-1 and its diffusion away from the source. The second equation describes the dynamics of the concentration of the JNK/AP-1 TF, produced at a basal rate and degraded linearly. Hill kinetics describe production of JNK/AP-1 induced by *egr* as well as inhibition of JNK/AP-1 by JAK [50]. The third equation describes the dynamics of the concentration of the JAK TF, produced at a basal rate and degraded linearly. Hill kinetics describe positive feedback dependent on Upd. Inhibition of JAK by JNK/AP-1 is described, and the model is tested with and without this mathematical term (highlighted in red) to check if this interaction exists within the system. The fourth equation describes the dynamics of the concentration of Upd, produced at a basal rate and degraded linearly. Hill kinetics describe the production of Upd induced by JNK/AP-1 and its diffusion [26,47,50].

Technically, the modeling was done based on a nondimensionalized version of the model, rescaled to reduce the number of free parameters. The system was then defined as follows:

$$\frac{dEIG(t,x)}{dt} = \gamma\left(1(x) - EIG + k_{act,EIG}\,\frac{JNK^{n1}}{K_{m,act,EIG}^{n1} + JNK^{n1}}\right) + \nabla^2 EIG$$

$$\frac{dJNK(t,x)}{dt} = \gamma\left(1 - k_{deg,JNK}\,JNK + k_{act,JNK}\,\frac{EIG^{n2}}{(K_{m,act,JNK}^{n2} + EIG^{n2})(1 + k_{inh,JNK,by,JAK}^{ni1}\,JAK^{ni1})}\right)$$

$$\frac{dJAK(t,x)}{dt} = \gamma(1 - k_{deg,JAK}\,JAK + k_{act,JAK}\,UPD\,\frac{JAK^{n3}}{(K_{m,act,JAK}^{n3} + JAK^{n3})(1 + k_{inh,JAK,by,JNK}^{ni2}\,JNK^{ni2})})$$

$$\frac{dUPD(t,x)}{dt} = \gamma\left(1 - k_{deg,UPD}\,UPD + k_{act,UPD}\frac{JNK^{n4}}{K_{m,act,UPD}^{n4} + JNK^{n4}}\right) + d\nabla^2 UPD$$

For each dimensionless parameter, the parameter dependencies and the scanned parameter range are represented in the Table 1 below:

We performed a global analysis by a sampling approach introduced in Rausenberger and colleagues [147]. Specifically, the parameter space is explored by drawing $10^6$ parameter sets uniformly on logarithmic scale within the interval from $10^{-3}$ to $10^2$. The Hill coefficients are drawn uniformly between 1 and 4. Parameter sets were simulated for each model (with and without repression) and under different conditions of diffusion–$D_{Upd} > D_{Egr}$, $D_{Upd} < D_{Egr}$ and $D_{Upd} = D_{Egr}$. The models were simulated on a 1D spatial domain x = [0, 1]. To represent the wound site, a high basal production of *egr* was fixed within a finite space at x = [0, 0.05], while basal rate of production of *egr* for the rest of the region was set to 0. The rate of production of Upd was defined as a function of JNK/AP-1 activity [16,18,43,50]. For all states u, initial conditions were set as u (t = 0,x) = 0.1.

The steady-state pattern of JAK/STAT and JNK/AP-1 gradients are defined as "observed" bistable pattern if the Pearson correlation coefficient to the experimental curve is greater than 0.7 for both gradients.

**Table 1. Parameters ranges and dependencies tested in the model.**

| Dimensionless parameter | Functional relation to systems parameter | Scanned parameter range |
|---|---|---|
| $\gamma$ | $\frac{k_{deg,EIG}\,L^2}{D_{EIG}}$ | 0.01…10,000 |
| $\hat{k}_{deg,JNK}$ | $\frac{k_{deg,JNK}}{k_{deg,EIG}}$ | 0.01…10,000 |
| $\hat{k}_{deg,JAK}$ | $\frac{k_{deg,JAK}}{k_{deg,EIG}}$ | 0.01…10,000 |
| $\hat{k}_{deg,UPD}$ | $\frac{k_{deg,UPD}}{k_{deg,EIG}}$ | 0.01…10,000 |
| $\hat{k}_{act,EIG}$ | $\frac{k_{act,EIG}}{b_{EIG}}$ | 0.00001…10 |
| $\hat{k}_{act,JNK}$ | $\frac{k_{act,JNK}}{b_{JNK}}$ | 0.01…10,000 |
| $\hat{k}_{act,JAK}$ | $\frac{k_{act,JAK}\,D_{EIG}}{L^2\,b_{UPD}\,b_{JAK}}$ | 0.01…10,000 |
| $\hat{k}_{act,UPD}$ | $\frac{k_{act,UPD}}{b_{UPD}}$ | 0.01…10,000 |
| $\hat{K}_{m,act,EIG}$ | $\frac{K_{m,act,EIG}\,D_{EIG}}{L^2\,b_{EIG}}$ | 0.00001…10 |
| $\hat{K}_{m,act,JNK}$ | $\frac{K_{m,act,JNK}\,D_{EIG}}{L^2\,b_{JNK}}$ | 0.01…10,000 |
| $\hat{K}_{m,act,JAK}$ | $\frac{K_{m,act,JAK}\,D_{EIG}}{L^2\,b_{JAK}}$ | 0.01…10,000 |
| $\hat{K}_{m,act,UPD}$ | $\frac{K_{m,act,UPD}\,D_{EIG}}{L^2\,b_{UPD}}$ | 0.01…10,000 |
| $\hat{k}_{inh,JAK,by,JNK}$ | $\frac{k_{inh,JAK,by,JNK}\,b_{JAK}L^2}{D_{EIG}}$ | 0.01…10,000 |
| $\hat{k}_{inh,JNK,by,JAK}$ | $\frac{k_{inh,JNK,by,JAK}\,b_{JNK}L^2}{D_{EIG}}$ | 0.01…10,000 |
| $\hat{n}_i$ | $n_i$ | 1…4 |
| $d$ | $\frac{D_{UPD}}{D_{EIG}}$ | 1.5…4 |

The pattern of JAK/STAT and JNK/AP-1 gradients are defined as "simple" bistable pattern if the steady states along the spatial axis fulfill the following features:

i. JAK at x = 0 lower than JAK at x = 1

ii. JNK/AP-1 at x = 0 higher than JNK/AP-1 at x = 1

iii. Relative difference of JAK higher than 10%

iv. Relative difference of JNK/AP-1 higher than 10%

## Supporting information

**S1 Fig. *egr*-mediated tissue damage induces JNK/AP-1 signaling, epithelial barrier dysfunction, and senescent-like cell cycle stalling in G2. (A)** Schematic of a central XY, YZ, and XZ section through a third instar wing imaginal disc epithelium. Different shades represent the pouch, hinge and notum domains. Dashed lines represent the anterior-posterior (A/P) and dorsal-ventral (D/V) compartment boundaries. The YZ section visualizes the pseudostratified monolayer organization of the disc proper, and the overlying peripodial cell layer. XZ section visualizes the apical to basal orientation of epithelial cells in the wing pouch. **(A')** Damage-induced JNK/AP-1 signaling mediates survival or apoptosis through control of cell cycle progression [29,36]. JNK also controls production of ROS, which is thought to act as signaling molecule but also induces oxidative damage, which could trigger p53-dependent apoptosis. However, p53 is activated by the G2/M kinase Cdk1 and thus competent to mediate damage-induced apoptosis in G1. **(B, C)** XY view of control **(B)** and *egr*-expressing **(C)** discs after 24 h of expression stained for filamentous Actin (phalloidin). XZ sections through the tissue were visualized along dotted yellow lines. Immunostaining for E-cadherin (E-Cad, magenta, adherens junction marker) and Discs large (Dlg, green, basal polarity marker) reveals reduced cell adhesion and cell polarity, and thus barrier integrity, in an *egr*-expressing disc when compared to a control disc. A total of *n* = 3, control and *n* = 3, *egr*-expressing discs were evaluated. **(D, E)** A control **(D)** and *egr*-expressing **(E)** discs after 24 h of expression assessed for senescence-associated beta galactosidase (SA-β-gal, cyan) activity using a SA-β-gal assay. Discs also express the JNK/AP-1 activity reporter *TRE-RFP* (magenta). Images show *TRE-RFP* overlay with the same discs as shown in Fig 1. A total of *n* = 23, 24 h control and *n* = 22, 24 h *egr*-expressing discs were evaluated from *N* = 2 independent experiments. **(F-I)** A time-course analysis of JNK/AP-1 reporter *TRE-RFP* (magenta) and EdU incorporation to detect DNA replication activity (EdU, Cyan) in control discs and *egr*-expressing discs. Dynamic reduction in S-phase activity and thus cycling cells is detected in the JNK/AP-1 signaling domain after 7 h **(F, G)** and 14 h **(H, I)** of *egr*-expression. A total of *n* = 3, 7 h; *n* = 4, 14 h control discs and *n* = 5, 7 h; *n* = 3, 14 h *egr*-expressing discs were evaluated. Maximum projections of multiple confocal sections are shown in **F-I**. Discs were stained with DAPI to visualize nuclei. Scale bars: 50 μm. (TIFF)

**S2 Fig. Pro-mitogenic Upd cytokines are up-regulated in the JNK/AP-1 signaling cells after *egr*-mediated tissue damage. (A-C)** A control **(A)** and *egr*-expressing **(B)** discs assessed for *upd-lacZ* cytokine reporter up-regulation after 24 h of expression. Graph **(C)** displays mean ± SEM for *n* = 7, control discs; *n* = 7, *egr*-expressing discs. Mann–Whitney *U* test was performed to test for statistical significance. **(D, E)** UMAP plots of control (top row) and *egr*-expressing (bottom row) discs analysed after 24 h of expression by single-cell RNA-Seq technology (see [72] for details). *Rn*-expression marks the *rn*-GAL4-positive cell population in control discs, and those that survive in *egr*-expressing discs. Induction of *upd1-3* transcripts in

the *rn*-positive cell population of *egr*-expressing discs (bottom row) can be clearly detected. Plots were generated using the Scope Wing Atlas [72]. Source data for quantifications provided in S1 File. Maximum projections of multiple confocal sections are shown in **A and B**. Discs were stained with DAPI to visualize nuclei. Scale bars: 50 μm.
(TIFF)

**S3 Fig. JAK/STAT signaling is activated de novo upon tissue damage and separates from JNK/AP1 signaling domains. (A)** Schematic showing dynamic patterning of the JAK/STAT signaling reporter (*Stat92E-dGFP*, green) activity in developing, undamaged wing imaginal discs. Note the gradual loss of hinge-specific patterns from mid to late third larval instar stages. **(B-D)** Undamaged control discs at 0 h (**B**), 7 h (**C**), and 14 h (**D**) of inductive temperature shift to 30˚C. Disc also express the JNK/AP-1 reporter *TRE-RFP* (magenta) and the dynamic JAK/STAT reporter *Stat92E-dGFP* (green and grey). **(H-J)** Wing discs after 0 h (**H**), 7 h (**I**), and 14 h (**J**) of *egr*-expression in the pouch using *rn*-GAL4. Disc also express the JNK/AP-1 reporter *TRE-RFP* (magenta) and the dynamic JAK/STAT reporter *Stat92E-dGFP* (green and grey). Increasing JNK/AP-1 reporter activity is seen in the pouch and increasing JAK/STAT reporter is seen in the pouch and hinge from 7 h to 14 h of *egr*-expression. **(E-G, K-M)** *Stat92E-dGFP* reporter fluorescence intensity, traced along the hinge of control and *egr*-expressing discs at 0 h (**E, K**), 7 h (**F, L**), and 14 h (**G, M**) of *egr*-expression. Colored arrow-heads indicate similar positions along hinge regions where JAK/STAT reporter activity is developmentally low but progressively increasing in *egr*-expressing discs (compare **G** and **M**). Also note that the average JAK/STAT reporter activity is higher at 14 h of *egr*-expression than in controls. Graphs display hinge traces from 1 representative disc. A total of $n = 7$, 0 h; $n = 12$, 7 h; and $n = 5$, 14 h control discs and $n = 7$, 0 h; $n = 22$, 7 h; and $n = 19$, 14 h *egr*-expressing discs were evaluated from $N = 2$ independent experiments. **(N, O)** Expression of *egr* using the *30A-GAL4* hinge driver. Discs were also expressing the *Stat92E-dGFP* (green) reporter and efficiency of JNK/AP1-activation was assessed by staining for the JNK-target MMP-1 (magenta). Please note how *egr*-expression by the (weak) 30A-GAL4 induces JAK/STAT activity in the expected expression domain but only where MMP-1 expression and thus JNK/AP1 signaling is low. Conversely, JAK/STAT activity is low where MMP-1 expression and thus JNK/AP1 signaling are high (yellow arrows). The yellow dashed line separates different z-sections on top and bottom. A total of $n = 5$, control and $n = 15$, *egr*-expressing discs were evaluated. **(P-R)** A time-course analysis of *TRE-RFP* (magenta) and *Stat92E-dGFP* (green) reporter activity in *rn*-GAL4 control discs after 7 h (**P**), 14 h (**Q**), and 24 h (**R**) of inductive temperature shift to 30˚C. **(S-W)** Quantification of the fluorescence intensity of the JNK-reporter *TRE-RFP* (**S, T**) and the JAK/STAT reporter *Stat92E-dGFP* (**U, V**) within the pouch ("pouch") and in a 15-μm band outside the pouch domain ("periphery") (**W**) in control wing disc (*rn*[ts]-GAL4) after 7 h, 14 h, and 24 h of inductive temperature shift to 30˚C. Graphs displays mean ± SEM for $n = 10$, 7 h control discs; $n = 16$, 14 h control discs; $n = 17$, 24 h control discs. One-way ANOVA with Holm–Sidak's multiple comparisons test was performed to test for statistical significance. Source data for quantifications provided in File S1. Maximum projections of multiple confocal sections are shown in **B-D and H-J.** Discs were stained with DAPI to visualize nuclei. Scale bars: 50 μm.
(PDF)

**S4 Fig. JNK/AP-1 represses JAK/STAT activity independent of pouch-specific transcription factors or cell death. (A, B)** *p35*-expressing control clones (red, **A**) and *p35,hep*[act]-coexpressing clones (JNK+, red, **B**) at 28 h after clone induction, stained for the pouch-specific transcription factor Nubbin (cyan). JNK/AP-1 signaling clones show no ectopic expression of Nubbin in hinge (yellow arrowheads); however, Nubbin is repressed in clones within the

pouch. A total of $n = 6$, control and $n = 6$ discs with clones coexpressing $p35,hep^{act}$ were evaluated. **(C)** $p35,hep^{act}$-coexpressing clones (red) in a disc expressing the Rn(E/F)-eGFP reporter (cyan) as a readout for the expression of the pouch-specific transcription factor Rotund (Rn) at 28 h after clone induction. JNK/AP-1 signaling clones show no ectopic clonal expression of Rn in the hinge domain (yellow arrowheads). A total of $n = 10$ discs were evaluated. **(D, E)** A control **(D)** and *egr,p35*-coexpressing disc **(E)** after 24 h of expression. Discs also express *TRE-RFP* (magenta) and *Stat92E-dGFP* (green) reporters. Note the absence of JAK/STAT reporter activity in the central (undead) JNK/AP-1 signaling cells. A total of $n = 17$, control discs and $n = 29$, *egr,p35*-coexpressing discs were evaluated from $N = 2$ independent experiments. **(F, G)** A control **(F)** and *scrib-RNAi*-coexpressing disc **(G)** after 44 h of expression in the central *rn*-GAL4 domain. Discs are also expressing the *TRE-RFP* (magenta) and *Stat92E-dGFP* (green) reporters. Note how JNK/AP-1 and JAK/STAT signaling cells in the targeted *rn*-GAL4 domain largely separate into distinct areas. A total of $n = 10$, control were evaluated from $N = 2$ independent experiments. A total of $n = 21$, *scrib-RNAi*-coexpressing discs were evaluated from $N = 3$ independent experiments. Discs were stained with DAPI to visualize nuclei. Scale bars: 50 μm.
(TIFF)

**S5 Fig. Mutual repression by JNK/AP-1 and JAK/STAT organizes a bistable signaling field. (A)** Counts of the correlation coefficients (binned) between the experimental curves of the JAK/STAT gradient shown in Fig 5B and the solutions obtained from the simulation for all simulated parameter sets in the unidirectional (blue) and the mutual repression (red) model. The mutual repression model results in substantially more solutions with positive correlations for the experimentally observed JAK/STAT gradient than the unidirectional repression model. Thus, the mutual repression model is better at recapitulating the response of JAK/STAT to the activation of JNK/AP-1. **(B)** Counts of "observed bistable" (red) and "not observed bistable" (blue) solutions derived from the mutual repression model examined by determining correlation coefficients between the experimentally and simulation-derived gradient curves ("observed" approach). Counts were plotted against the width of the JNK gradient (spatial position of half of maximum JNK activity) in the simulated steady-state field. No bistable solutions can be found when the JNK-gradient is very narrow, i.e., the correlation coefficients of these solutions with the experimental data is low. **(C)** Distribution of "bistable" and "not bistable" results obtained by examining the simulated solutions of the mutual repression model using "observed" (correlation coefficient-based) and "simple" (descriptive) criteria. Distribution of all simulated solutions is plotted depending on their differences between minimum and maximum JAK/STAT and JNK activity at steady state in the spatial field. Each point represents a simulation result for a different parameter set. Color code as in **(D-H)**. **(D)** Subsets of the simulation results shown in **(C)** classified as "observed bistable." **(E)** Subset of the simulation results shown in **(C)** classified as "simple bistable." **(F)** Subset of the simulation results shown in **(C)** classified as fulfilling both "simple and observed" criteria of bistability. **(G)** Subset of the simulation results shown in **(C)** classified as not bistable by "simple" (descriptive) criteria. **(H)** Subset of the simulation results shown in **(C)** classified as not bistable by "observed" (correlation coefficient) criteria. **(G')** Representative steady-state simulation results for JNK/AP-1 (magenta) and JAK/STAT (green) signaling gradients classified as "not bistable" patterns. **(G")** Simulated spatiotemporal evolution of JNK/AP-1 and JAK/STAT activation for the same examples shown in **(G')**. These plots represent kymographs with a space and time axis, where pathway activation is plotted from low (blue) to high (green) activity. **(I)** For parameter combinations classified as "simple bistable" patterns, there is no relationship between the width of the Eiger gradient (spatial position of half of the maximum concentration) and the

spatial position of the intersection point between JNK/AP-1 and JAK/STAT curves. **(J)** For parameter combinations classified as "simple bistable" patterns, there is a strong relationship between the width of the JNK activity gradient (spatial position of half of the maximum activation) and the spatial position of the intersection point between JNK/AP-1 and JAK/STAT curves. **(K)** For parameter combinations classified as "simple bistable" patterns, there is no relationship between the width of the Eiger gradient (spatial position of half of the maximum concentration) and the width of the JNK activity gradient (spatial position of half of the maximum activation). The Eiger gradient is rather narrow in "simple bistable" solutions, while the JNK gradient covers the whole range for different parameter combinations. **(L)** Numerical space of 19 parameters simulated in the mutual repression model, and influence of the parameters' numerical values on scoring positive as a "simple bistable" solution. Distribution of solutions classified as "bistable" using "simple" (descriptive) criteria for selection from the mutual repression model. Each solution is plotted as an event on density plots against the values of all 19 simulated parameters within each simulated set. Parameters represent reaction rate constants—for activation, degradation, and inhibition of simulated components, as well as hill coefficients for different regulators of the model. The bar graphs on the diagonal show the frequency of "simple bistable" classifications within the evaluated parameter ranges. The following parameters preferably lead to a simple bistable pattern: large $k_{act}$ JNK values (i.e., high activation rates of JNK), but also small $K_m$ JAK. Source data for graphs (**A**-**K**) derived from mathematical modeling provided in File S3. Source data for graph (**K**) derived from mathematical modeling provided in File S4.
(PDF)

**S6 Fig. Regulation of JAK/STAT activation in JNK/AP-1 signaling cells. (A-C)** Test for genetic complementation of the LOF mutants $dome^{G0441}$, $hop^{9p5}$, and $stat^{85C9}$ (grey bars) by GFP-tagged Dome, Hop, and Stat92E fosmid lines (purple bars) in adult female (**F**) and male (**M**) flies (see Experimental procedures). Dome-GFP is unable to rescue male lethality in $dome^{G0441}$ LOF background to the expected rate of rescue, i.e., 50% of adults in F1 (red dashed line), $n = 2$ independent replicates (**A**). Hop-GFP successfully rescues male lethality in $hop^{9p5}$ LOF background from 0% to 47% of viable adults, at the expected rate of rescue in F1 (red dashed line), $n = 3$ independent replicates (**B**). Stat92E-GFP partially rescues larval lethality in $stat^{85c9}$ null mutants from 0% to 15% pupariation, at approximately half the expected rate of a full rescue (red dashed line), $n = 3$ independent replicates (**C**). Thus, the Hop-GFP and Stat92E-GFP fosmid lines were used for further analysis of protein localization in $egr$-expressing discs. **(D)** Western blots analyzed for Stat92E-GFP fusion protein expression in imaginal discs. HP-1-GFP genotypes were included as positive control, along with a non-GFP-expressing negative control from wild-type imaginal disc extracts. Increasing concentrations were loaded and probed with anti-GFP and anti-Tubulin antibody. The GFP-tagged Stat92E protein isoforms are detected at the expected MW (125 Kda and 110 Kda) and likely representing 4 overlapping isoforms running in 2 separate weight ranges (71.2–76.8 kD and 85.6–92.8 kD plus GFP-tag). **(E)** A control $w^{118}$ disc used as a negative control to determine anti-GFP antibody background. **(F, F')** A control (**F**) and $egr$-expressing disc (**F'**) after 24 h of expression also ubiquitously expressing a GFP-tagged Hop protein. A total of $n = 5$ control and $n = 4$ $egr$-expressing discs were evaluated from $N = 2$ independent experiments. **(G, G')** A control (**G**) and $egr$-expressing disc (**G'**) after 24 h of expression also ubiquitously expressing the GFP-tagged Stat92E protein. While Stat92E expression is low in the medial hinge folds of control discs, the hinge shows elevated expression in egr discs (purple asterisk). This is consistent with JAK/STAT activity inducing expression of Stat92E to promote self-activation. In contrast, Stat92E-GFP protein levels remains low in the pouch (yellow asterisk). A total of $n = 8$ control and $n = 9$ $egr$-

expressing discs were evaluated from *N* = 2 independent experiments. **(H, H')** A control pouch **(H)** and a larger domain of *p35,egr*-coexpressing cells in the pouch **(H')** were used to test for changes in nuclear versus cytoplasmic localization of Stat92E-GFP within high JNK/AP-1-signaling cells. **(I)** Stat92E-GFP fluorescence intensity within nuclear and cytoplasmic area fractions of the JNK/AP-1 signaling pouch, calculated for *n* = 12 independent regions from control and *n* = 16 independent regions from *egr,p35*-coexpressing discs. The pouch domain in *egr,p35*-coexpressing discs fails to exhibit an increase in nuclear translocation of the *Stat92E*-GFP protein, indicative of basal levels of JAK/STAT signaling in these cells similar to controls. The surrounding hinge domain recapitulated elevated JAK/STAT signaling by elevated Stat92E-GFP protein expression and nuclear translocation (see Fig S6H and S6H'). **(J, J')** A control **(J)** and *egr*-expressing disc **(J')**, also expressing GFP-tagged Su(var)2-10/dPIAS and stained for GFP. A total of *n* = 5 control and *n* = 11 *egr*-expressing discs were evaluated. **(K, K')** A control **(K)** and *egr*-expressing disc **(K')** stained for Apontic (Apt). Yellow outline in **(K')** marks nonspecific tracheal staining. A total of *n* = 5 control and *n* = 7 *egr*-expressing discs were evaluated. **(L, L')** A control **(L)** and *egr*-expressing disc **(L')**, also expressing *ken-LacZ* and stained for anti-β-Galactosidase. A total of *n* = 7 control and *n* = 8 *egr*-expressing discs were evaluated from *N* = 2 independent experiments. **(M-O")** *RNAi*-expressing control discs **(M, N, O)**, *egr*-expressing **(M', N', O')** and *egr,RNAi*-coexpressing discs **(M", N", O")** visualized for *Stat92E-dGFP* reporter activity. *egr,ken-RNAi* **(M")** and *egr,su(var)2-10-RNAi* **(N")** coexpressing discs were dissected after 24 h of expression. *egr,apt-RNAi* coexpressing discs **(O")** were dissected even after 48 h of *egr* expression to promote efficiency of the RNAi-mediated knock-down. Note how knock-down of these negative regulators does not lead to increase in the JAK/STAT reporter activity in the central pouch domain where JNK/AP-1 signaling is expected to be high. A total of *n* = 12 *ken-RNAi*-expressing discs; *n* = 18 *egr-RNAi*-expressing and *n* = 26 *egr,ken-RNAi* coexpressing discs were evaluated from *N* = 2 independent experiments. A total of *n* = 21 *su(var)2-10-RNAi*-expressing, *n* = 10 *egr-RNAi*-expressing, and *n* = 30 *egr,su(var)2-10-RNAi* coexpressing discs were evaluated. A total of *n* = 12 *apt-RNAi*-expressing were evaluated from *N* = 2 independent experiments. A total of *n* = 20 *egr-RNAi*-expressing and *n* = 33 *egr,apt-RNAi* coexpressing discs were evaluated from *N* = 3 independent experiments. **(P, Q)** UMAP plots of control **(P)** and *egr*-expressing **(Q)** discs analysed after 24 h of expression by single-cell RNA-Seq technology (see [72] for details). *Rn*-expression marks the *rn*-GAL4-positive cell population in control discs **(P),** and those that survive in *egr*-expressing discs **(Q)**. Induction of *Ptp61F* and *Socs36E* transcripts in the *rn*-positive cell population of *egr*-expressing discs (bottom row) can be detected. Plots were generated using the Scope Wing Atlas [72]. **(R-T)** Expression of *UAS-Ptp61F-RNAi* (BDSC:56510) and *UAS-RFP* (magenta) using the *en-GAL4* driver **(S)** and expression of *UAS-Socs36E-RNAi* (BDSC:35036) and *UAS-RFP* (magenta) using the *en-GAL4* driver **(T)** leads to a distinct increase in developmentally patterned *Stat92E-dGFP* reporter (green) activity in the posterior wing hinge (yellow arrowheads) compared to stage-matched controls **(R)** dissected on D7 AED. **(U-W)** Quantification of changes to the development pattern of JAK/STAT activation, using fluorescence intensity measurements for the *Stat92E-dGFP* reporter. Black dashed lines in schematic **(U)** show regions in the anterior (A) and posterior (P) wing hinge that were analysed in control and *RNAi*-overexpressing ($en>RNAi^{OE}$) discs. Graph **(V)** displays mean ± SEM for *n* = 12, anterior and posterior regions in control discs; and *n* = 12, anterior and posterior regions in *en>RFP,Ptp61F-RNAi*-coexpressing discs. Paired *t* test was performed to test for statistical significance. Graph **(W)** displays mean ± SEM for *n* = 3, anterior and posterior regions in control discs; and *n* = 6, anterior and posterior regions in *en>RFP,Socs36E-RNAi*-coexpressing discs. Paired *t* test was performed to test for statistical significance. **(X, Y)** A control **(X)** and *UAS-STAT92E-3xHA* expressing disc, under the control of the *rn-GAL4* driver **(Y)**, stained for HA

(magenta). Initiating 12 h of expression on D7 AED leads to strong up-regulation of the *Stat92E-dGFP* reporter (green) in the wing pouch. Younger discs (after just 12 h of expression on D7 AED) have higher levels of developmentally regulated *upds*. Therefore JAK/STAT activation is easily detectable in these samples (when compared to after 24 h of expression in Fig 6J). Source data for quantifications provided in **S1 File**. Original western blot images provided in **S1 Raw Image File.** Maximum projections of multiple confocal sections are shown in **G-H'.** Discs were stained with DAPI to visualize nuclei. Scale bars: 50 μm.
(PDF)

**S7 Fig. Coactivation of JNK/AP-1 and JAK/STAT signaling causes escape from G2 arrest and apoptosis (A-D).** A control wing disc **(A)**, *Stat92E*-expressing **(B)**, *egr*-expressing **(C),** and *egr*,*Stat92E*-coexpressing disc **(D)**, which also express *UAS-GFP* (green), all under the control of *rn-GAL4*. Discs were stained for cleaved Dcp-1 (magenta) to visualize apoptosis. Note how almost all apoptotic debris is labelled by GFP **(D)**, demonstrating that apoptotic cells truly originate from *egr*,*Stat92E*-coexpressing cells. A total of $n = 9$, control discs; $n = 10$, *Stat92E*-expressing discs; $n = 11$, *egr*-expressing discs; and $n = 18$, *egr*,*Stat92E*-coexpressing discs were evaluated from $N = 2$ independent experiments. **(E, F)** A control **(E)** and *Stat92E*-expressing wing disc **(F)** at R0, also expressing the FUCCI reporter. A total of $n = 20$, control discs were evaluated from $N = 4$ independent experiments. A total of $n = 17$, *Stat92E*-expressing discs were evaluated from $N = 2$ independent experiments. **(G, H)** A *gfp*-expressing control **(G)** and *Stat92E*-expressing disc **(H)** at R0, assayed for S-phase activity by EdU incorporation. Elevated EdU is detected in the pouch domain ectopically expressing *Stat92E*. However, no changes were observed in the FUCCI cell cycle profile. However, no changes were observed in the FUCCI cell cycle profile. A total of $n = 10$, control discs and $n = 10$, *Stat92E*-expressing discs were evaluated. **(I)** Adult wings developing from of UAS-*Ptp61F*-RNAi expressing; *egr*-expressing and *egr*,*Ptp61F-RNAi*-coexpressing discs after 24 h of expression were scored according to wing size and morphology, as previously described [29,50]. Graphs display mean of binned wing scores emerging from $n = 44$, *Ptp61F-RNAi*-expressing; $n = 260$ *egr*-expressing and $n = 117$, *egr*,*Ptp61F*-RNAi-expressing discs from $N = 4$ independent experiments. Source data for quantifications provided in S1 File. Maximum projections of multiple confocal sections are shown in **E and F.** Discs were stained with DAPI to visualize nuclei. Scale bars: 50 μm.
(TIFF)

**S8 Fig. Tumors display bistable separation of JNK/AP-1 and JAK/STAT signaling domains and respective G2-phase and proliferative cell behaviors. (A, B)** A control wing disc **(A)** and wing discs expressing *scrib-RNAi* **(B)** for 44 h stained for E-cadherin (Ecad, adherens junction marker) (magenta) and Discs-large (Dlg, marker of basal polarity) (green) reveals loss of cell adhesion and cell polarity, and thus loss of barrier integrity (yellow arrowheads in **B**) in the *scrib-RNAi*-expressing pouch domain when compared to its respective hinge control domain (white arrowheads in **B**), as well as to the undamaged control pouch (yellow arrowhead in **A**). The pouch-hinge interface is marked with white dashed lines. A total of $n = 3$, control and $n = 3$ *scrib-RNAi*-expressing discs were evaluated. **(C-F)** A control wing disc **(C)** and wing discs expressing *scrib-RNAi* **(D)**, *Ras^{v12}*,*scrib-RNAi* **(E),** and *egr* **(F)** for 44 h. Discs also express the JNK/AP-1 reporter *TRE-RFP* (magenta) and JAK/STAT reporter *Stat92E-dGFP* (green). *TRE-RFP* and *Stat92E-dGFP* reporter fluorescence intensities were adjusted to subsaturation in *egr*-expressing discs and all genotypes were imaged at comparable settings; however, brightness settings were raised for all samples equally to prepare the figure. A total of $n = 5$, control; $n = 4$, *scrib-RNAi*-expressing; $n = 6$, *Ras^{v12}*,*scrib-RNAi*-expressing; and $n = 7$ *egr*-expressing discs were evaluated. **(G)** Schematic illustrates *TRE-RFP*-positive (JNK+, magenta) and

*TRE-RFP*-negative (JNK−, white) regions segmented from images for measurements. Graph represents *TRE-RFP* reporter fluorescence intensity measured within selected JNK+ and JNK− image masks. Graphs represent mean ± SEM for *n* = 6, *Ras^{v12}*,*scrib-RNAi*-expressing discs. Paired *t* test was performed to test for statistical significance. **(H)** A control disc expressing the *Stat92E-dGFP* (green) reporter and G2-specific FUCCI reporter *ubi-mRFP-NLS-CycB1-266* (magenta). A total of *n* = 6, control discs were evaluated. Source data for quantifications provided in S1 File. Discs were stained with DAPI to visualize nuclei. Scale bars: 50 μm. (TIFF)

**S1 Table. Key resource tables.** Reagents and genetic lines used in this study. (DOCX)

**S2 Table. Detailed genotypes.** Detailed genotypes listed per figure panel. (DOCX)

**S1 File. Source data for all experimental work.** Source data for all quantifications derived from experimental work presented in this study (Figs 1–8). (XLSX)

**S2 File. Source data for mathematical modeling Fig 5.** Source data for graphs derived from mathematical modeling presented in Fig 5. (XLSX)

**S3 File. Source data for mathematical modeling S5A–S5K Fig.** Source data for graphs derived from mathematical modeling presented in S5A–S5K Fig. (XLSX)

**S4 File. Source data for mathematical modeling S5L Fig.** Source data for graphs derived from mathematical modeling presented in S5L Fig. (XLSX)

**S1 Raw Images. Source data for western blots in S6 Fig.** Original western blot images for data shown in S6D Fig. (PDF)

## Acknowledgments

We thank the staff of the Life Imaging Center (LIC) in the Hilde Mangold House (HMH) of the Albert-Ludwigs-University of Freiburg for help with their confocal microscopy resources, and the excellent support in image recording. We specifically thank the DFG for supporting our imaging work through project number 414136422. We thank Erika Bach, David Bilder, Dirk Bohmann, Stephen Cohen, Fernando Diaz-Benjumea, Iswar Hariharan, Norbert Perrimon, Helena Richardson, Reinhardt Schuh, Y. Henry Sun, Mirka Uhlirova, and Pelin Volkan for sharing reagents and the Bloomington *Drosophila* Stock Center (BDSC), the Vienna *Drosophila* Stock Collection (VDRC), the University of Zurich ORFeome Project (FlyORF), the Developmental Studies Hybridoma Bank (DSHB), and the Monoclonal Antibody Core Facility at the Helmholtz Zentrum Munich for providing fly stocks and antibodies. We thank Florian Steinberg (University of Freiburg) for support with protocol, reagents, and imaging equipment for western blot assays. We also thank the SGBM graduate school for supporting our students.

## Author Contributions

**Conceptualization:** Janhvi Jaiswal, Anne-Kathrin Classen.

**Data curation:** Janhvi Jaiswal, Janine Egert, Anne-Kathrin Classen.

**Formal analysis:** Janhvi Jaiswal, Janine Egert, Raphael Engesser, Andrea Armengol Peyrotón, Liyne Nogay, Carlo Crucianelli, Anne-Kathrin Classen.

**Funding acquisition:** Anne-Kathrin Classen.

**Investigation:** Janhvi Jaiswal, Janine Egert, Raphael Engesser, Andrea Armengol Peyrotón, Liyne Nogay, Carlo Crucianelli, Anne-Kathrin Classen.

**Methodology:** Janhvi Jaiswal, Janine Egert, Raphael Engesser, Andrea Armengol Peyrotón, Vanessa Weichselberger, Isabelle Grass, Anne-Kathrin Classen.

**Project administration:** Isabelle Grass, Anne-Kathrin Classen.

**Supervision:** Clemens Kreutz, Jens Timmer, Anne-Kathrin Classen.

**Validation:** Janhvi Jaiswal, Andrea Armengol Peyrotón, Isabelle Grass, Anne-Kathrin Classen.

**Visualization:** Janhvi Jaiswal, Janine Egert, Liyne Nogay.

**Writing – original draft:** Janhvi Jaiswal, Raphael Engesser, Anne-Kathrin Classen.

**Writing – review & editing:** Janine Egert, Raphael Engesser, Liyne Nogay, Vanessa Weichselberger, Clemens Kreutz, Jens Timmer, Anne-Kathrin Classen.

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
