## [Editor Report · Decision Letter 0]

4 May 2022

Dear Dr Classen, 

Thank you for submitting your manuscript entitled "Mutual repression between JNK/AP-1 and JAK/STAT stratifies cell behaviors during tissue regeneration" for consideration as a Research Article by PLOS Biology.

Your manuscript has now been evaluated by the PLOS Biology editorial staff as well as by an academic editor with relevant expertise and I am writing to let you know that we would like to send your submission out for external peer review.

Once your full submission is complete, your paper will undergo a series of checks in preparation for peer review. Once your manuscript has passed the checks it will be sent out for review. To provide the metadata for your submission, please Login to Editorial Manager (https://www.editorialmanager.com/pbiology) within two working days, i.e. by May 06 2022 11:59PM.

If your manuscript has been previously reviewed at another journal, PLOS Biology is willing to work with those reviews in order to avoid re-starting the process. Submission of the previous reviews is entirely optional and our ability to use them effectively will depend on the willingness of the previous journal to confirm the content of the reports and share the reviewer identities. Please note that we reserve the right to invite additional reviewers if we consider that additional/independent reviewers are needed, although we aim to avoid this as far as possible. In our experience, working with previous reviews does save time. 

If you would like to send previous reviewer reports to us, please email me at ialvarez-garcia@plos.org to let me know, including the name of the previous journal and the manuscript ID the study was given, as well as attaching a point-by-point response to reviewers that details how you have or plan to address the reviewers' concerns. 

Kind regards,

Ines

--

Ines Alvarez-Garcia, PhD

Senior Editor

PLOS Biology

---

## [Decision Letter · Decision Letter 1]

29 Jun 2022

Dear Dr Classen,

Thank you for your patience while your manuscript entitled "Mutual repression between JNK/AP-1 and JAK/STAT stratifies cell behaviors during tissue regeneration" was peer-reviewed at PLOS Biology. Please also accept my apologies for the delay in providing you with our decision. Your manuscript has been evaluated by the PLOS Biology editors, an Academic Editor with relevant expertise, and by four independent reviewers.

The reviews are attached below. As you will see, the reviewers find the work potentially interesting, however they also raise a substantial number of concerns that would need to be addressed for us to consider further the manuscript for publication. Their concerns include data presentation, method explanation, several controls and more balanced data interpretation. Based on their specific comments and following discussion with the Academic Editor, it is clear that a substantial amount of work would be required, however, given the interest in the study, we would like to invite a comprehensive revision that addresses the reviewers’ comments.

Considering the extent of revision that would be needed, we cannot make a decision about publication until we have seen the revised manuscript and your response to the reviewers' comments. Your revised manuscript would need to be seen by the reviewers again, but please note that we would not engage them unless their main concerns have been addressed. 

We appreciate that these requests represent a great deal of extra work, and we are willing to relax our standard revision time to allow you 6 months to revise your study. Please email us (plosbiology@plos.org) if you have any questions or concerns, or envision needing a (short) extension.

**IMPORTANT - SUBMITTING YOUR REVISION**

3. Resubmission Checklist

a) *PLOS Data Policy*

b) *Published Peer Review*

Sincerely,

Ines

--

Ines Alvarez-Garcia, PhD

Senior Editor

PLOS Biology

Reviewers' comments

Rev. 1:

The manuscript by Classen and colleagues describes a molecular mechanism underlying the segregation of cell behaviors during regeneration. They start with their and others' observations that cells engaged in JNK/AP1 signaling, identified with TRE-RFP and generally stalled in G2, segregate to different areas of the Drosophila wing disc than cells engaged in JAK/STAT signaling identified with STAT92E-GFP. Because JNK signaling cells transcribe the Upd ligands that activate JAK/STAT, they ask why aren't JNK cells also positive for STAT92E? They find that cells begin expressing both reporters, but over 24h of JNK activation, they lose JAK/STAT signaling. They next develop a mathematical model to reproduce the JNK-induced activation of JAK/STAT non-autonomously while simultaneously inhibiting JAK/STAT autonomously. The model suggests that mutual repression of JNK by JAK/STAT and of JAK/STAT by JNK is sufficient to explain the results, so they next look for the mechanisms underlying the repression. Using co-expression and gene knockdown, they report that the JAK-STAT induced transcription factor Zfh2 inhibits the JAK/STAT pathway; and the phosphatase Ptp61F, which is upregulated by JNK, inhibits the JAK/STAT downstream transcription factor, Stat92E. Returning to cell behaviors, they report that when cells are forced to activate both pathways, apoptosis increases as the G2 arrest is lost and these cells now enter the cell cycle. They conclude by examining a genetically induced tumor of the wing disc and showing that JAK/STAT and JNK signaling segregate in this tissue also, as they do in the non-tumorous wing discs.

The conclusion of the study is that JNK signaling associated with wounds is sufficient to induce two segregated populations of cells with different signaling pathways and different cell behaviors. This conclusion is interesting, novel, and identifies a mechanism to address the important question about how wounds pattern the cell responses around them. However, the paper is very hard to read, and the data are hard to evaluate because they are poorly presented. Further, there is a lot of over-interpretation; re-writing sections with a lighter tough, acknowledging incomplete answers, is required.

About the paper being hard to read: First, it is in a strange style, with the figures rigidly forming the sections of the results, as though it were a poster. The text and the figure legends are each, and sometimes both together, insufficient to understand the figures. The order of the figures is also hard to make sense of, compounded by the frequent references to supplementary figures.

About the data: it is difficult to evaluate how much reproducibility and variability have been assessed. A great number of figure panels in this study show a single example, and it's not clear how to interpret the assurance in the methods that n is at least two for every experiment when it's not addressed in the text, figure, or legend. For example, the only data addressing reproducibility and variability in Fig. 1 is Fig. I,J; even panels M,N measure only a single sample. Further, 1I and J are difficult to understand and represent mostly background levels - why not remove background before assessing the levels of the reporters? The same is true for all of Fig. 2, for Fig 5 I,J,O,P; and most of the extensive supplemental data (n=2 is not enough for statistics, as done in Suppl Fig. 1.2). Further, there is no indication that samples were scored blinded. This is a particular concern when there is so much uniqueness in how image data is manipulated - sometimes a section, sometimes a projection, sometimes thresholded, despeckled, denoised, etc. Such individualized image analysis for each experiment can open the door to unconscious bias, which is why blinded analysis is so important.

Some examples of over-interpretation:

*Lines 148-150, "We show that upd cytokines activate JAK/STAT only non-autonomously in a cell population spatially segregated from high JNK-signaling and upd-producing cells." The authors do not show that upd cytokines are even synthesized and secreted, let alone that they activate JAK/STAT. Although it is a fair inference, it is not "shown".

*In lines 186-188, "Altogether, these experiments demonstrate that tissue damage associated with high levels of JNK/AP-1 signaling repress JAK/STAT activity in a cell autonomous manner, independent of developmental competence." It is clear that the overexpressed levels of JNK/AP1 are repressing JAK/STAT, but not that it's the "tissue damage".

* In Fig. S3, the difference in MMP1 levels +/- Zfh2 may be statistically significant, but it is certainly not a large effect. The Y-axis is misleading as it does not start at 0 and so exaggerates the effect. This small effect cannot support the strong statements in the abstract or discussion about how JAK/STAT suppress JNK activation via Zfh2 - tone it down.

* The main takeaway of the manuscript is that JNK signaling can self-organize the patterning of cell behaviors, G2-stall or proliferation, each mediated by its own pathway. However, the only place that proliferation is quantified with respect to Jak/Stat signaling is in tumors, Fig.6I. Although the effect may be significant, it is not large (perhaps a mean of 2.3 to a mean of 2.9).

*In line 391, there is no evidence here that segregation of signaling networks provides an advantage to tumors.

Two final questions:

In line 298/Fig 4J, co-expressing Stat92E with egr does not demonstrate that dephosphorylation of Stat92E is rate limiting, only that Stat92E is rate limiting. Can you restore normal reporter levels by co-expressing ptp61F along with egr and stat93E-HA?

Is it possible the G2 stall cells are endocycling? Their nuclei look significantly bigger than surrounding nuclei.

Small things:

1. Please reserve the ">" (greater than) symbol for Gal4-driven gene expression, and do not use it for "TRE>RFP" and "STAT92E>GFP". Instead use "TRE-RFP" and "STAT92E-GFP". Otherwise, it appears you are using rnGal4 to drive Eiger and TREGal4 to drive RFP and STAT92Egal4 to drive GFP, all at the same time.

2. The 2° antibodies used for the immunofluorescence experiments are not described.

3. Show rn>GFP in Fig. 1, for readers not familiar with this domain.

4. Could the authors double-check that the DAPI image in 1A is not (accidentally) the same disc shown in 1C?

5. The authors should cite the Park et al NCB 1997 (Greco) paper showing spatial segregation of cell behaviors around wounds.

6. What are the diagonal yellow arrows in Fig. 1?

7. Line 167, referring to Fig. S2A seems to be an error.

Rev. 2:

Jaiswal and colleagues report mutually exclusive activities of JNK and JAK/STAT during regeneration and tumor growth in Drosophila wing discs. Both of these pathways are activated in the context of tissue damage in a number of different experimental systems, and are required for regeneration. This study addresses their spatial and temporal organization. Overexpression of Eiger (Drosophila TNF) activates JNK cell autonomously and ,JAK/STAT non-autonomously. This group has shown in a 2019 eLIFE paper that cells that activate JNK arrest in G2 and this arrest provides a pro-survival benefit. The current manuscript reports that activation of JNK and STAT are mutually exclusive such that STAT is not activated in cells with high JNK activity. Depletion of a phosphatase encoded by ptp61F in cells with active JNK leads to STAT activity in these cells, leading the authors to conclude that ptp61F is the reason for the lack of STAT activity. Mutual exclusion is observed also with ectopic activation of JNK with a constitutively active kinase, which is sufficient to inhibit STAT activity even in cells that normally shows high developmental STAT activity. A similar spatial relationship is seen in tumors driven by a combination of oncogenic RasV12 and the loss of polarity through scrib mutations. Forced co-activation of JNK and STAT resulted in cell death. The authors propose that self-organized, mutually exclusive expression of JNK and JAK/STAT allows JNK-active cells to survive and STAT-active cells to proliferate during regeneration and in tumor growth.

JNK and JAK/STAT are conserved signaling pathways with clear importance for development, tissue regeneration and cancer. Understanding how they corporate in tissue regeneration or in tumor growth is a significant goal. A self-organizing and mutually exclusive arrangement of JNK and JAK/STAT activities is of potential interest to readers of PLoS Biology. I have several reservations about the approach and data interpretation that should be addressed first, however. For example:

General concern 1. The authors use signal area in many of their quantifications (e.g., Fig. 5D). This is a usual practice for wing discs and works well because they made up of single cell layers. The problem here is that experimental manipulations used changed tissue organization to result in 3D folds (e.g., Fig. 5B). How do the authors justify using 2D areas to quantify signals in tissue that is clearly folded in 3-dimension?

General concern 2. Overexpression of eiger in the pouch is a standard technique used to ablate the pouch. That means the disc regions being analyzed are a mix of living and dead cells whereas control discs will have only living cells. So changes observed could be an indirect consequence of cells dying. To give a specific example, Fig. 1 shows convincingly that JNK and dynamic STAT reporter activities are mutually exclusive and that EdU incorporation and STAT reporter activity are reduced in JNK-active cells. Could that be a simple consequence of JNK-active cells dying? Similarly, an alternate explanation for Fig 2G-J is that as JNK activity in the pouch increases, cells start to die and no longer incorporate EdU. Knowing what fraction of the pouch is made of live/dead cells at R0 and focusing the analysis only to live cells could address these concerns.

Specific concerns and recommendations:

1. Fig. 2A-C provides the most convincing evidence that cells with ectopic JNK activity repress STAT within themselves but activate STAT in the neighbors. These cells are also 'undead' due to co-expression of p35. Is it possible to make undead cells without using JNK and ask if you see a similar effect on STAT? That would be a nice control.

2. Fig. 2F. I cannot tell how the A/P boundary determined here. If JNK is being activated in the P compartment and JNK-high cells are supposed to inhibit STAT, why is STAT-GFP showing up here? I see a lot of pyknotic (small condensed) nuclei in the supposed A compartment even though JNK is activated only in the P compartment. To demonstrate across-compartment activation, it would help to see the whole disc with the compartments clearly defined.

3. Fig. 2K-M. First, the DAPI panels are showing STAT>dGFP. In these, GFP in the pouch seems to first increase from K to L then decrease from L to M. Again, this could be an indirect consequence of pouch cells dying.

4. Fig. 4. The increase in STAT>GFP reporter from B to C is not impressive. It is also unclear how the authors know to compare the area just within the yellow circle. They should first identify the JNK-active area in an unbiased manner and then make the comparison. If they want to make the conclusion that the mechanism by which JNK represses STAT is through ptp61F (as shown in Fig. 7C), they need to connect JNK to ptp61F in their experimental system rather than cite the work of others that address this in other contexts (ref 23, 116). Without such data, the authors cannot rule out the possibility that ptp61F simply sets a threshold for STAT activity and that JNK acts through another target to overcome it. And an alternate explanation for Fig. 4I that shows increased STAT>dGFP is that it is due to elevated overall STAT from STAT-HA expression, not that extra STAT is saturating Ptp61F (line 299-300). Can the authors rule this out?

5. Fig. 5. The nuclei in the boxed region in F are larger than those in the box in E and I wonder if the authors are looking at the peripordial cells instead of the columnar pouch. Again, 3D folding is apparent in these discs and could be confounding the analysis. The cell cycle data could also be interpreted in more than one way. For example, adding stat92E to eiger increased the fraction of G1 cells, decreased the fraction of G2 cells, and increased EdU+ nuclei. The authors interpret this as stat92E pushing the cells out of a (protective) G2 arrest by JNK. But the data are equally consistent with stat92E stalling cells in G1 and prolonging S, without changing G2 (because what is being measured are % and rations that are relative and not absolute). Cell doubling times or mitotic index will be needed to distinguish between these possibilities.

6. The authors conclude throughout the manuscript that cells with high JNK activity experience a 'a senescent-like G2 cell cycle stall' (line 118); 'their cell cycle is senescently arrested (line 37); acquire senescent phenotypes' (line 452). Yet, there are no data shown to demonstrate senescence such as staining for senescence markers.

7. There are no data shown to support the conclusion or even address the possibility that '…cells with activated JNK/STAT suppress JNK activation via Zfh2' (line 30-31, Fig. 3B, Fig 7C legend).

8. Loss of cells to death seems to be missing in the modeling in Fig. 3. Cell death could make a difference if sustained JNK activity is needed to keep activating JAK/STAT. Loss of JNK active cells to death could cause a decline in JAK/STAT activity with time, making the unidirectional repression appear like bidirectional repression in Fig. 3D.

Rev. 3:

In this manuscript, entitled "Mutual repression between JNK/AP-1 and JAK/STAT stratifies cell behaviors during tissue regeneration", Jaiswal and co-authors investigate the pattern and function of JNK/AP-1 and JAK/STAT activation in wound healing/regeneration. To this end, they use a wound healing model system in the fly wing imaginal disc. In this system, they over-express the Tumor Necrosis Factor TNF-α/Eiger, which, if I understand correctly, activates cell death through JNK/AP-1. At the same, this event leads to the activation of JNK in the center of the disc, which also protect those cells from apoptosis. Conversely, JNK active cells (center of the disc) express the JAK/STAT activator Upds, which activates JAK/STAT at the wing periphery, but not in those central cells. The authors conclude that high JNK/AP-1 signalling locally inhibits JAK/STAT in those central cells. Overall, the authors propose a feedback system between the JNK/AP-1 and JAK-STAT pathways; this system exhibits bistability and leads to compartmentalization of the activity of these two pathways. They propose that such compartmentalization favours regeneration by segregating proliferating cells to the disc edge. In addition, compartmentalization prevents the activation of JNK/AP-1 and JAK/STAT in the same cell, which would lead to excess senescence. Finally, the authors associate disrupted JNK/AP-1 and JAK/STAT patterns with the induction of senescence and proliferation in chronic tissue damage and senescence models, still in the wing disc.

In my view, this work poses interesting questions in a powerful model system. The authors make a wise use of fly genetics to dissect the feedbacks between JNK/AP-1 and JAK/STAT. Here, modeling is fundamental to sort out the behaviour of such a bistable signalling network. Despite these positive premises, I find that the model system is not introduced clearly enough, which could be problematic considering the broad readership of PLOS Biology. The manuscript should also be improved in the presentation and use of the mathematical model. Most importantly, I feel that the conclusions of this work could be made stronger if the authors had used their time-courses of JNK/AP-1 and JAK/STAT activation to investigate the compartmentalization model.

Specific comments

Fig. 1

1) The authors over-express TNF-α/Eiger to induce wound healing and regeneration. I am not too familiar with this system and I have a hard time understanding how it works. I thought that over-expression of TNF-α/Eiger induces cell ablation in the center of the disc, through activation of JNK/AP-1. However, I gather from the introduction that JNK/AP-1 protect cells from cell death, although it activates inflammation pathways. Thus, I do not understand if Eiger over-expression induced cell death, protects cells from it or both at different times. I invite the authors to describe their model system more extensively, in particular clarifying what is the injury, what cellular processes occur during wound healing and which ones occurs during regeneration. Along similar lines, I invite the authors to provide legends within the Figures to guide the reader through the various reporters and perturbations that are used in this manuscript.

Fig. 2

2) The authors obtain time-courses of the patterning of JNK/AP-1 and JAK/STAT activation. I think that the presented results are interesting, as they show regions of transient co-activation of the two pathways. However, I am surprised that the authors did not analyse the implications of these temporal dynamics when building their regulatory network (see below). Can these dynamics be used to discriminate between different models and select parameters?

Fig. 3

The authors present in Fig. 3 a model of JNK/AP-1 and JAK/STAT compartmentalization. They compare two models: a "uni-directional repression model" and a "mutual repression model". The "mutual repression model" is a simple bistable switch and there are no surprises regarding its behaviour. Instead, the "uni-directional repression model" undergoes patterning because eiger expression is pre-patterned (bEIG(x)). Thus, the qualitative implications of the model are quite straightforward.

The authors use an "unbiased approach" to ask in which regions of the parameter space bistable patterns are generated. This approach is not described in detail. In addition, simulation results are not shown, so that we do not know how patterning occurs, how patterns differ for different parameter choices and what is considered a bistable pattern or not. The authors present a score table to show how often "bistable patterns" occur when varying different parameters (Fig. S3D), but I am not sure what to learn from the plot. I suggest the authors to better present and improve their modeling efforts as follows:

3) The authors should show simulation results (dynamics and final pattern), for different parameter choices. They should show representative patterns that are classified as bistable and not.

4) The authors should describe the procedure used to explore the parameter space and score results.

5) The authors should compare the time evolution of JNK/AP-1 and JAK/STAT activation predicted in silico with the one obtained experimentally.

6) The authors describe two qualitatively different patterns: the "simple bistable pattern" and the "observed bistable pattern". I do not think that "the observed bistable pattern" is a better fit of the experimental curves shown in Fig. 3A. For example, the authors point at JAK/STAT activation decreasing at the right boundary, but they do not emphasize JNK exhibiting larger variations on the left. I invite the authors to use a fitting or smoothening procedure to obtain an experimentally observed reference pattern; then, they would compare their model results to that experimental curve, without the need of postulating an alternative "simple bistable system".

7) The authors should investigate what is the importance of the gradient of eiger production bEIG(x). Is that needed for bistability? Is the feedback system simply amplifying the differences between the domains that have high and low bEIG(x)?

8) In the paragraph starting at line 257, the authors say "the length scale-independent space utilized by the model raises the possibility that this mutual repression network could act at length scales ranging from neighboring cells to multicellular tissues." I think that this sentence is misleading, as the system has various characteristic lengths that can be obtained by taking the square root of any diffusion constant divided by any degradation rate. Unless those parameters change accordingly, the model is not scale-independent.

Figure 5

9) The authors show that the activation of JAK/STAT in cells having active JNK/AP-1 leads to excess apoptosis. Do the authors think that this phenotype is cell-autonomous or does it result from the perturbation of the spatial pattern of JNK/AP-1 and/or JAK/STAT? Can the authors achieve an experimental condition in which JAK/STAT is active at the center of the disc and JNK/AP-1 at its periphery? Does this altered pattern impair cellular behaviour and regeneration?

Rev. 4:

In this interesting article, Jaiswal et al investigate the complex signalling pathways activated following epithelial tissue disruption that ensure distinct cell behaviours occur in mutually exclusive tissue domains during tissue regeneration. In particular, through a combination of in vivo genetics, imaging and mathematical modelling, they uncover factors that drive mutual repression between JNK and JAK/STAT signalling in damaged Drosophila wing discs. Given that there is considerable interest in elucidating the signalling mechanisms that lead to effective tissue regeneration and how these differ in chronic non-healing wounds, this study is interesting and timely. This is generally a well written paper and the results presented are intriguing, however I would recommend this manuscript for publication as an Article in PloS Biology only if the following points are addressed.

Major comments

1. In my opinion the main limitation of this paper in its current form is the lack of comprehensive description of the model system being used. The authors present the Eiger over-expression wing disc model as a 'extensively used' model to study tissue regeneration following damage. To understand the implications of (and interpretation of) data in this manuscript, the Introduction would benefit from more background on the model being studied.

For example, does Eiger over-expression drive loss (i.e. death) of some wing disc cells or is the Eiger-induced 'damage' not lethal? In the introduction they write "We recently reported that high JNK/AP-1 signaling facilitates survival in wounds and tumors by mediating a cell cycle stall in G2 which is characterized by anti-apoptotic and senescent features [31]". Yet they see upregulation of Dcp1 in the JNK domain in Figure 1B (which seems a paradox). Given this Dcp1 staining it indeed appears after 24h that cell death is occurring throughout most of the JNK domain but this is not directly mentioned in the accompanying text. This would be useful information to understand how similar this model is to a more traditional 'wound'. If JNK+ cells do eventually die, it would also be informative to include a timeline of this death (over 7h, 14h and 24h) and an indication of how long these cells will continue secreting Upd?

2. Related to this, in the current Eiger model, is there an 'end point' that can be analysed for a readout of successful regeneration? i.e. if cells in the JNK+ region do eventually die, can one look if surrounding STAT+ cells totally repopulate the area? Perhaps adult wings could be examined to reinforce the claims around the importance of these mutually exclusive domains? Later on, they could manipulate the JNK/JAK-STAT domains (e.g. using ptp61f RNAi) and show images of the adult wings (the phosphatase RNAi should fail regeneration and adult wings should look abnormal)?

3. Related to point 1 above, is it known whether the mutual repression observed between JNK and JAK-STAT also occurs when the wing discs are damaged using an alternative method e.g. by forcing ectopic rpr expression (and does this also depend on expression of Ptp61F)?

4. Have the authors tried making clones overexpressing Upd ligands to determine if this also induces similar JAK-STAT activity in surrounding cells? In other words, is secretion of Upds on their own sufficient to establish these mutually exclusive domains, or is JNK activation forcing JAK/STAT activation in neighbouring cells by other means?

5. It would be interesting if the authors could comment on how far the Upd ligands are thought to travel from secreting cells in the JNK active domain to activate STAT. It appears that STAT is activated throughout most of the wing disc perhaps suggesting that Upds travel far? Or perhaps STAT cells participate in a positive feedback loop and make their own Upd which activates STAT in neighbouring cells? Positive feedback is included in the presented model, but it is unclear whether this is thought to be cell autonomous or non-cell autonomous amplification?

6. Figure 3: It would be interesting (although not essential for this revision) if the authors could use the mathematical model to further explore how the size of clone or the duration/intensity of JNK affects the signalling dynamics and mutual repression.

7. For the Ptp61F knockdown experiments in Figure 4, it is important that the authors can demonstrate that the same phenotype can be achieved with multiple independent RNAi lines to preclude off target effects. Ideally the authors would also show (although it is not essential) the effectiveness of the RNAi knockdown (e.g. using RT-qPCR).

8. To fully describe the system, it is important to analyse the expression of Ptp61F. For example, can in situ hybridisation be used to show i) that Ptp61F is induced only in the JNK compartment and ii) explore the timeline of Ptp61F induction (in control vs Egr expressing discs). Alternatively, two EGFP-FlAsH-StrepII-TEV-3xFlag tagged Ptp61F lines are available from Bloomington which could reveal Otp61F levels.

`

9. The data presented demonstrates that loss of Ptp61F can elevate STAT activity, but is over-expression Ptp61F also sufficient to stop activation of JAK STAT? A UAS-Ptp61F transgenic line is available from Bloomington so the authors should be able to perform this experiment .

10. In the experiments to demonstrate that Ptp61F inhibits JAK-STAT activity via dephosphorylation of STAT in which STAT92 is over-expressed to saturate the Ptp61F dephosphorylase activity, is it possible that STAT92 over-expression would increase JAK-STAT activity (in the presence of egr) even if unrelated to Ptp61F? Indeed it appears that as rn>stat92E alone causes a modest increase of JAK/STAT signalling. To confirm the role of Ptp61F, can transgenic modified STAT constructs be used such as one in which STAT cannot be dephosphorylated? Alternatively, if this experiment is genetically too difficult to perform, perehaps the authors could expose the discs to Tyrosine phosphatase inhibitors?

11. Figure 6: Does Ptp61F expression downstream of JNK signalling also repress STAT activity in the tumour model?

12. The model presented in Figure 7 is interesting but would benefit from a more thorough description of when 'transient or low level' JNK levels might occur. Given that in the eiger overexpression wing disc model it takes 24h for full mutual suppression between JNK and STAT, it will be interesting to know how a more transient burst of JNK activity affects STAT?

Minor comments

1. In Figure 2A-B, the authors use expression of p35 to prevent cell death in the 'hep-active' discs. It is a little difficult to observe the details of the clones (and surrounding STAT activity) - the Figure would be improved by inclusion of higher magnification insets. It could also be beneficial to include an additional control here - staining in wild type wing discs where there is no ectopic p35 expression.

2. Figure 2K - is this STAT activity not DAPI? And if not, please show low mag images of STAT throughout wing disc over time.

3. Please include more detail on previous work with Zfh2 - for example, what are the consequences of RNAi Zfh2 in JAKSTAT domain and elevation of JNK?

4. Figure S2K - please explain in more detail how the region of interest was quantified?

5. Please show evidence MMP1 upregulated by JNK

6. In Figure 4 F-J, it is unclear why the discs were examined at 12h post induction when all other experiments use 7h, 14 and 24h.

7. In Figure 5, there appears to be considerable apoptosis (via Dcp1 staining) in egr expressing discs and these spread across most of the disc when STAT92 is also expressed. Could the authors suggest how these cells (originating from the JNK+ domain) might spread so far across disc?

---

## [Decision Letter · Decision Letter 2]

24 Feb 2023

Dear Dr Classen,

Thank you for your patience while we considered your revised manuscript entitled "Mutual repression between JNK/AP-1 and JAK/STAT stratifies senescent and proliferative cell behaviors during tissue regeneration" for consideration as a Research Article at PLOS Biology. Your revised study has now been evaluated by the PLOS Biology editors, the Academic Editor and the four original reviewers. 

In light of the reviews (attached below), we are pleased to offer you the opportunity to address the remaining points from the reviewers in a revision that we anticipate should not take you very long. We will then assess your revised manuscript and your response to the reviewers' comments with our Academic Editor aiming to avoid further rounds of peer-review, although might need to consult with the reviewers, depending on the nature of the revisions.

**IMPORTANT - SUBMITTING YOUR REVISION**

3. Resubmission Checklist

a) *PLOS Data Policy*

b) *Published Peer Review*

Sincerely,

Ines

--

Ines Alvarez-Garcia, PhD

Senior Editor

PLOS Biology

Reviewers' comments

Rev. 1: Andrea Page-McCaw – note that this reviewer has signed her review

The authors have fully addressed my comments and concerns. The bistable self-organizing patterning of JNK and JAK/STAT is an interesting and important finding, and this paper is suitable for publication.

Rev. 2:

Jaiswal and colleagues report mutually exclusive activities of JNK and JAK/STAT during regeneration and tumor growth in Drosophila wing discs. The revision addresses my concerns from the first round of review except for the last comment (specific concern #8) that the authors asked clarification for. I will try to provide that here.

In figure 3 of the original manuscript and figure 5 of the revised version, the authors show two different models. In the first model, JNK activates JAK/STAT via Upds but then JAK/STAT represses JNK (unidirectional repression). In the second, initial activation of JAK/STAT is followed by later repression of JAK/STAT by JNK (bidirectional repression). My concern is that even if unidirectional model is the correct one, as JNK-active cells die, they cannot produce Upds to activate JAK/STAT, leading to loss of JAK/STAT activation. This would then look as if initial activation of JAK/STAT is followed by repression of JAK/STAT (mutual repression model) when what is happening is not repression but loss of activation. The modeling does not include complications due to the death of JNK-active, Upd-producing cells over time. Can this be ruled out? I suspect that response to my General concern 2 (the role of cell death in JAK/STAT activation) may already have addressed this specific concern.

Rev. 3:

The authors have significantly revised and improved their manuscript. They provide a more detailed description of the experimental system, of the model and of their strategy of exploration of the parameter space. They also provide representative plots of the temporal dynamics of their model and additional analysis.

They have decided to not compare the time-evolution of their model with the experimental results, which I think is a missed opportunity. They refer to the fact that "while we have a fairly good understanding of the behavior of our reporters, we did not vigorously determine their (linear) activity ranges, so it would be naive to correlate fluorescence levels with precise mathematical activities (at this point in time)." On the other hand, they do use those fluorescence levels to analyze spatial patterns, so they are quite confident on those reporters. Nevertheless, the scope of the manuscript is authors' choice. I still request the authors to quantify the pattern of JNK and JAK/STAT activation as a function of distance from the center at different time-points, using the data reported in Fig.1 and Fig. 3, unless they have already done it and I missed it.

I do not think that they have addressed my last question on the effects of activation of JAK/STAT at the center of the disc and JNK/AP-1 at its periphery on apoptosis. They have commented on the reciprocal effects of JNK and JAK/STAT, but not on its effect on apoptosis. I request the authors to comment on the effects of this perturbation.

I think that, once implemented these minor revisions, the manuscript will be acceptable for publication.

Rev. 4:

We thank the authors for the additional experiments/quantification provided in this revised version of the manuscript. Whilst the authors have provided additional data to address some of our original comments and explained why some of the additional experiments we had requested are not feasible, we still have the following concerns that we feel should be addressed (and should be relatively straight forward to address) before final publication:

1. Regarding levels of Ptp61F:

We requested more information on the expression of Ptp61F, including the timing of expression and whether expression is restricted to the egr-expressing (JNK positive) domain. The authors refer to scRNA-seq data to help address this question (supplemental figure S6.3). However in these scRNAseq graphs, it appears as if egr expression actually induces Ptp61F (and the new additional candidate Socs36E) throughout the disc, rather than just in the rn-expressing domain. Some additional quantification using this scRNAseq data would be important to confirm how the expression of these key factors change following egr induction (within and outside the rn positive domain). These new scRNAseq data also suggest that Ptp61F and Socs36E induction might restrain JAKSTAT signalling throughout the disc, rather than in just the JNK positive domain? A comment on this would be useful.

2. Are the JNK positive cells quiescent or senescent:

The authors state that JNK positive cells undergo senescence. However, they now also provide additional data to show that these JNK cells actually re-enter the cell cycle and re-populate a significant portion of the disc after JNK activity subsides (G-trace experiments). This raises the question whether these cells are senescent or whether they are rather becoming temporarily quiescent?

3. Image quantification:

As pointed out by another Reviewer there were some concerns with quantification in the original manuscript. Related to point (1) above, given the paper's core concept surrounds the establishment of 2 separate signalling domains, there is a lack of quantification of key proteins (including levels of upd, ptp16F and socs36E) in these 2 separate domains (within the same egr expressing discs). The majority of quantification shows that levels of upd, ptp61f etc change over time following egr induction, but this is compared with control discs. What is missing is the quantification of the spatial distribution of these proteins in the 2 domains following egr induction. Moreover in Figure 3M: it is unclear where the measurements of Jak/STAT92 activation have been taken. In the main text, the authors refer to 2 subdomains within the disc (pouch and hinge) but it is unclear whether the quantifications of JAK/STAT intensity reported in the graph come from the pouch, the hinge or both. As over the course of 24h the levels changes in the two domains, maybe it would be better to show the two quantifications separately rather than showing a cumulative values?

---

## [Editor Report · Decision Letter 3]

25 Mar 2023

Dear Dr Classen,

Thank you for your patience while we considered your revised manuscript entitled "Mutual repression between JNK/AP-1 and JAK/STAT stratifies senescent and proliferative cell behaviors during tissue regeneration" for publication as a Research Article at PLOS Biology. This revised version of your manuscript has been evaluated by the PLOS Biology editors and the Academic Editor.

Based on our Academic Editor's assessment of your revision, we are likely to accept this manuscript for publication, provided you satisfactorily address the data and other policy-related requests stated below.

We expect to receive your revised manuscript within two weeks. 

*Published Peer Review History*

*Press*

Sincerely,

Ines

--

Ines Alvarez-Garcia, PhD

Senior Editor

PLOS Biology

DATA POLICY:

Many thanks for submitting the data underlying the graphs shown in the figures. I have checked them all and I am missing the data for the following figures:

Fig. 3C, F, G; Fig. 5B-D, B’-D’, B’’-D’’; Fig. S3.1E-G, K-M; Fig. S5.1A-G, G’, G’’, H-K and Fig. S5.2

In addition, please ensure that figure legends in your manuscript include information on WHERE THE UNDERLYING DATA can be found, and ensure your supplemental data file/s has a legend.

We require the original, uncropped and minimally adjusted images supporting all blot and gel results reported in an article's figures or Supporting Information files. We will require these files before a manuscript can be accepted so please prepare and upload them now. Please carefully read our guidelines for how to prepare and upload this data: https://journals.plos.org/plosbiology/s/figures#loc-blot-and-gel-reporting-requirements

---

## [Editor Report · Decision Letter 4]

14 Apr 2023

Dear Dr Classen,

Thank you for the submission of your revised Research Article entitled "Mutual repression between JNK/AP-1 and JAK/STAT stratifies senescent and proliferative cell behaviors during tissue regeneration" for publication in PLOS Biology. On behalf of my colleagues and the Academic Editor, Philipp Niethammer, I am delighted to say that we can in principle accept your manuscript for publication, provided you address any remaining formatting and reporting issues. These will be detailed in an email you should receive within 2-3 business days from our colleagues in the journal operations team; no action is required from you until then. Please note that we will not be able to formally accept your manuscript and schedule it for publication until you have completed any requested changes.

PRESS

Sincerely, 

Ines

--

Ines Alvarez-Garcia, PhD

Senior Editor

PLOS Biology
